# Therapeutic and Protective Effects of Liposomal Encapsulation of Astaxanthin in Mice with Alcoholic Liver Fibrosis

**DOI:** 10.3390/ijms20164057

**Published:** 2019-08-20

**Authors:** Yu Chiuan Wu, Han Hsiang Huang, Yi Jhen Wu, Ioannis Manousakas, Chin Chang Yang, Shyh Ming Kuo

**Affiliations:** 1Hualien Armed Forces General Hospital, Hualien County 97144, Taiwan; 2National Kaohsiung University of Hospitality and Tourism, Kaohsiung City 81271, Taiwan; 3Department of Veterinary Medicine, National Chiayi University, Chiayi City 60054, Taiwan; 4Department of Biomedical Engineering, I-Shou University, Kaohsiung City 82445, Taiwan; 5Bioptik Technology, INC, Miaoli County 36045, Taiwan

**Keywords:** astaxanthin, liposome, alcohol, liver fibrosis, intraperitoneal injection

## Abstract

Astaxanthin (Asta) has been demonstrated to possess anti-inflammatory, antitumor, and free radical-clearing activities. However, the poor stability and low water solubility of Asta hamper its bioavailability. The objectives of this study were to fabricate Asta-loaded liposomes (Asta-lipo) and investigate the therapeutic effects of Asta-lipo on alcoholic liver fibrosis in mice. The mice were administered with Asta-lipo or liposomes alone prior to a 3-week dose containing 30% alcohol with or without feeding with a second dose of 30% alcohol. The prepared Asta-lipo of 225.0 ± 58.3 nm in diameter, had an encapsulation efficiency of 98%. A slow release profile of 16.2% Asta from Asta-lipo was observed after a 24-h incubation. Restorative actions against alcoholic liver fibrosis were observed after oral administration of Asta-lipo for 4 weeks. Hepatic repair, followed by a second dose of 30% alcohol, suggested that Asta-lipo exerted protective and reparative effects against liver injuries induced by repeated consumption of alcohol. The changes of serum ALT and AST values were principally in consistence with the histopathologic findings. Asta-lipo exerted rapid and direct effects against repeated alcohol-induced liver disease, whereas Asta-lipo given orally could boost recovery from liver injuries obtained due to previous long-term alcohol use. These data demonstrate that Asta-lipo has applicable protective and therapeutic potential to treat alcohol-induced liver diseases.

## 1. Introduction

Alcoholic liver disease (ALD) is a complex condition that includes a wide range of conditions from simple fatty liver (steatosis) to more severe liver injuries, such as alcoholic hepatitis, cirrhosis, and hepatocellular carcinoma. Excess alcohol consumption is the primary cause of global liver-related mortality [1]. Lesions associated with this condition may occur separately, simultaneously, or sequentially in patients. Alcoholic fatty liver is the first hepatic response to binge drinking or chronic alcohol consumption. The alcohol-induced fatty liver might recover after alcohol withdrawal; however, more severe liver injuries, such as liver inflammation and hepatitis or steatohepatitis, may persist even after alcohol withdrawal. In-progress hepatic diseases may result in the development of liver fibrosis, which is characterized by excessive scarring, vascular alterations, and functional failure of the liver [2]. In general, recovery from alcohol-induced liver injury after cessation of alcohol consumption is associated with disease severity [3].

Astaxanthin (Asta) is a xanthophyll carotenoid sourced from various marine creatures, such as shrimp, salmon, and the algae *Haematococcus pluvialis*. Asta has been demonstrated to exhibit numerous biological activities, including anti-inflammatory, antitumor, and free radical-clearing actions [4,5]. Over the past decade, several studies have investigated the preventive or therapeutic effects of Asta against various disorders or diseases, such as cancer, age-related macular degeneration, inflammation, and cardiovascular oxidative stress, and have examined its effects for the promotion of immune responses [6]. Asta has also been found to be capable of preventing diet-induced obesity and hepatic steatosis in vivo, as evidenced by its stronger preventive and nonalcoholic steatohepatitis (NASH) therapeutic effects than those exerted by vitamin E on a mouse model [7,8]. Asta is unstable; it drastically decomposes on exposure to heat, light, and oxygen; and is poorly water soluble and dispersible in aqueous solutions. These characteristics cause pure Asta compounds to exhibit low bioavailability, which limits their pharmaceutical or biomedical applications. Therefore, biomaterial manipulations, such as encapsulating Asta into a hydrophilic polymer matrix or incorporating Asta into liposomes by using emulsion or suspensions, have been investigated and developed to efficiently improve the solubility and stability of Asta.

Liposomes, which comprise a colloidal and vesicular structure, are composed of one or more lipid bilayers [9]. They have been widely used as drug carriers for peptides/proteins, hormones, antibiotics, and anticancer agents in the drug delivery system. Liposomes are biocompatible, biodegradable, and nontoxic, and their structure is similar to that of cell membranes. Thus, liposomes have become a suitable alternative for drug vehicles to increase the stability of encapsulated agents and improve the pharmacodynamics or therapeutic index of drugs [10,11]. Among liposome classes, phosphatidylcholine liposomes are the most widely used because their phospholipid components are similar to the relevant components of the cell membrane. The phospholipid 1,2-distearoyl-*sn*-glycero-3-phosphocholine (DSPC) has been used for liposome preparation for years; we have successfully fabricated DSPC liposome-encapsulated propranolol, which stimulated human osteoblastic cell proliferation and differentiation as effectively as did pure propranolol [12]. Furthermore, the prepared liposomal propranolol significantly increased bone volume density of the tibia and spine in ovariectomized rats (OVX rats) at a low dose of one-tenth to 1% of the therapeutic range of propranolol, thus demonstrating that the established liposome encapsulation can increase the bioavailability and potency of the target compound in vivo [13]. Thus, DSPC-based liposomes were employed as carriers for low bioavailable and unstable Asta in the current study. The encapsulation efficiency, drug release, and cytotoxic properties of Asta-encapsulated liposomes (Asta-lipo) were assessed. The improvement in steatohepatitis and liver fibrosis induced by alcohol intake was further evaluated through two administration routes—intraperitoneal injection (i.p.) and oral administration (Per os, p.o)—on a mouse model. Alanine aminotransferase (ALT) is known as serum glutamate pyruvate transaminase. ALT is primarily produced in the liver, and its activity is much lower in other tissues; thus, ALT is specifically measured to assess liver injury or hepatic diseases. Elevated ALT levels are highly associated with the presence of damaged liver cells in liver injury and hepatitis. Aspartate aminotransferase (AST), or originally called serum glutamic oxaloacetic transaminase, is present in the mitochondria and cytoplasm of liver cells. When liver cells are damaged, AST is released. AST is also present in other tissues such as the heart, skeletal muscle, and kidney. The combined measurements of these two liver enzyme activities are essential for the diagnosis and assessment of liver disease or injury. Therefore, the preventive and reparative effects of Asta-lipo were examined and verified using histopathology and serum ALT and AST analyses in this study.

## 2. Results

### 2.1. Asta Encapsulation and In Vitro Release of Asta from Asta-Lipo

The procedures of Asta-lipo preparation are shown in Figure 1A. A TEM observation revealed that the produced liposomes were spherical with a diameter of 225.0 ± 58.3 nm (mean ± SD, Figure 1A). The encapsulation efficiency of Asta-lipo was 98%, and approximately 16.2% of Asta was released from the liposome-encapsulated Asta after 24-h incubation under 40-rpm shaking (Figure 1B). A low and slow release profile of Asta from liposomes was observed, which is probably attributed to the low solubility and precipitation of Asta in the aqueous solution. Thus, a relatively low release rate of Asta was detected through high-performance liquid chromatography (HPLC) in 24 h.

### 2.2. Assessment of Effects on Cell Viability and Cell Cycle

The effects of Asta-lipo and pure liposomes on the cell viability of L929 fibroblasts are shown in Figure 2. A significant decrease (*P* < 0.05) in the viability of L929 fibroblasts was observed after the addition of 0.01 mg/mL of pure liposomes for 24 h. No obvious inhibition in L929 fibroblast viability was observed after treatment with 0.01 mg/mL of Asta-lipo. These findings indicate the nontoxic properties of Asta-lipo in normal cells, which rationalized the liposomal encapsulation of Asta in the current experimental design. The effects of Asta-lipo and liposomes on the cell cycle of L929 fibroblasts were determined through flow cytometry. Treatment with Asta-lipo or pure liposomes (0.01 mg/mL) for 24 h did not significantly affect the cell cycle of L929 fibroblasts (Figure 3 and Table 1). Additionally, a 48-h exposure to Asta-lipo or pure liposomes clearly did not influence the cell cycle of L929 fibroblasts, demonstrating that the addition of Asta-lipo or pure liposomes barely interferes with the mitosis and growth of normal cells.

### 2.3. Measurement of Animal Weight

The weight of mice in the experimental groups was measured on alternate days throughout the in vivo tests. The weight of most mice gradually increased in Groups C–J, which received Asta-lipo for 2-week/4-week treatment periods. Mice in Groups O and P at week 2 as well as Groups Q and R at week 4 showed significant weight loss compared with the normal group A. Given the identical dosage form and treatment period, except for the 4-week treatment with Asta-lipo, mice that received oral administration presented higher weight gain than did those treated with intraperitoneal injection (Figure 4A,B). By contrast, the untreated mice (Group B) lost approximately 23% body weight by day 21. The data indicated that mice treated with Asta-lipo intraperitoneally or orally in the absence (Groups C–F) and presence of a subsequent oral treatment of 30% ethanol (Groups G–J) gradually gained body weight or more stably maintained body weight. By contrast, mice that received pure liposomes orally or intraperitoneally gradually lost weight, particularly those in the interspersed treatment with 30% ethanol oral administration groups (Groups O–R). In addition, a slight decrease in body weight at week 2 was observed in some Asta-lipo administered groups (Groups D, G and H) compared to that without any treatment (Group S). The possible reason is a short adaptive period due to the extra intake or injection of Asta-lipo. This might result in a slight decrease in body weight (Figure 4A). However, the body weight in pure liposomes treated groups in both modes (Groups M, N, Q and R) was still significantly heavier than that of the mice with induced alcoholic liver disease but without treatment at week 4 (Group T, Figure 4B). The reason could be that the constituents of pure liposomes are similar to a high-fat diet and therefore administration of pure liposomes led to an increase in body weight.

### 2.4. Gross Pathological Observation of Mouse Livers Treated with Asta-Lipo and Pure Liposomes

Figure 5 shows the gross pathological observations of the mouse livers in each experimental group at weeks 2 and 4. Group B comprised livers from mice fed with 30% ethanol for 21 days, which showed obviously enlarged liver lobes and several white deposits of fat or fibrotic tissues (positive control). The mice livers of Groups E and F, in which mice received alcohol for 4 weeks orally or intraperitoneally with Asta-lipo treatment, exhibited fewer enlarged liver lobes and/or fewer white deposits of fat or fibrotic tissues; moreover, the shape of the liver lobes in both the groups was largely restored. By contrast, obvious enlarged liver lobes and some fibrotic tissues (white arrows) were still seen in Groups C and D, which received the treatments for shorter periods. Similarly, in Groups G and H (treatment on day 1 and 30% ethanol intake on day 2), enlarged liver lobes and some liver lobe areas with white deposits of fat or fibrotic tissues were observed. The shape of liver lobes was obviously restored, and fewer enlarged/swollen liver lobes were seen in Groups I and J. The liver in Groups O and Q, in which mice received pure liposomes, exhibited enlarged liver lobes and several areas with white deposits as well as few fibrotic tissues (yellow arrows), indicating no ameliorative or reparative effects of pure liposomes on liver injury and fibrosis.

### 2.5. Histopathological Analysis of Mouse Livers Treated with Asta Liposomes or Pure Liposomes

The preventive and reparative effects of Asta-lipo on alcoholic liver fibrosis and injury in vivo were further investigated and evaluated through histopathological analysis. Figure 6A showed the H&E staining of normal mice liver. Mice that received 30% ethanol orally for 21 days exhibited a large amount of inflammatory cell infiltration, hepatocyte degeneration and necrosis, and clear fibrotic tissues in the liver at low magnification (Group B, Figure 6B). Neutrophilia was distinguished in the areas of inflammatory cell infiltration at high magnification. The appearance of infiltration of inflammatory cells and fatty degeneration-like areas with hepatocyte degeneration and necrosis and deposition of fibrous tissues, were clearly observed, indicating successful induction of alcoholic liver disease. Additionally, the liver tissues of mice in Group S, which received no treatment prior to the 2-week induction of alcoholic liver disease, exhibited irregularly accumulated and disorderly arranged hepatocytes. On week 4 after the induction without treatments, the liver tissues showed an obvious decrease in fibrotic tissues and improvement in hepatocyte necrosis. However, the swelling and degeneration of hepatocytes and fatty degeneration-like hepatic areas were still noted (Group T) compared with observations in Group S and Group B, implicating the progression of hepatic self-repairing effects. In the livers of mice in which alcoholic liver disease was induced through oral treatment with Asta-lipo for 2 weeks (Group C), the amount of fibrotic tissues and hepatocyte necrosis decreased, whereas inflammatory cell infiltration was still observed. Additionally, several fatty degeneration-like areas remained in the liver (Group C, Figure 6C). However, in the livers of mice that received alcohol intake for 3 weeks and intraperitoneal treatment with Asta-lipo, clear hepatocyte swelling and inflammatory cell infiltration were observed (Group D, Figure 6D), suggesting progression to acute inflammation to some extent. The histopathological differences between Groups C and D were consistent with their ALT levels, with the ALT level in Group D being slightly higher than that in Group C. In the group which oral Asta-lipo was administered for 4 weeks, obvious restoration of hepatic tissues was observed. Normal hepatic structures, such as the central vein, hepatocytes, and sinusoids, were found, despite slight inflammatory cell infiltration (Figure 6E). By contrast, the livers of mice induced with alcoholic liver disease that received intraperitoneal injection of Asta-lipo for 4 weeks (Group F, Figure 6F) showed some normal hepatic structures coupled with fewer fatty degeneration-like areas, less hepatocyte swelling, or less inflammatory cell infiltration compared with the groups that received Asta-lipo via both routes for 2 weeks (Groups C and D).

The reparative effects of Asta-lipo on alcoholic liver disease plus the second-dose alcohol consumption (30% ethanol) were observed and are shown in Figure 6G–6J. Notably, both intraperitoneal and oral administration of Asta-lipo for 2 and 4 weeks exerted restorative actions against existing alcoholic liver disease plus repeated alcohol consumption. However, obvious inflammatory cell infiltration and swollen hepatocytes and more fibrotic tissues were found in the livers of mice that were given Asta-lipo orally for 2 weeks (Group G) compared with in livers of those that were given Asta-lipo intraperitoneally for 2 weeks (Group H). The histopathological features suggested that short-term (2 weeks) oral administration of Asta-lipo was unable to fully restore liver injury caused by alcoholic liver disease with subsequent alcohol consumption (Group G), but intraperitoneal Asta-lipo administration exerts stronger restorative and preventive effects on the status of repeated alcohol intake-induced liver injuries (Group H). The significantly decreased collagen found in Masson’s Trichrome staining in Groups C, D, G and H compared to that in Group S or T (Figure 6S–T), is suggesting that despite observation of hepatocyte swelling and inflammatory cell infiltration in H&E staining, 2-week Asta-lipo administration seems to have some effects against hepatic fibrosis in alcoholic liver disease. Moreover, lower collagen content can be found in Groups E, F and H (Figure 7), demonstrating stronger remission of liver fibrosis in these groups. These data are consistent with the results by H&E staining showing that intraperitoneal Asta-lipo administration exerts stronger restorative and preventive effects on the status of repeated alcohol intake-induced liver injuries.

### 2.6. Analysis of Sera ALT and AST Levels

ALT measurement is a relatively accurate indicator of liver injury or liver diseases because ALT is mainly produced in the liver [14]. The activity of ALT is much lower in other tissues. AST is present in the mitochondria of liver cells and is released when liver cells die. Other tissues such as the heart, muscle, kidney, and brain tissues also produce AST. Serum ALT and AST levels were further examined to validate the hepatic function in mice in each experimental group. The results revealed that ALT levels in Groups C, E, and F, but not in Group D, returned within the normal range, indicating that the oral administration of Asta-lipo could recover liver function during both shorter and longer treatment periods prior to 3-weeks of 30% alcohol feeding in mice. Among groups with the addition of interspersed treatment with 30% alcohol (Figure 8A), ALT levels in Groups H, I, and J, but not in Group G, dropped within the normal range, suggesting that intraperitoneal administration of Asta-lipo for 2 and 4 weeks improved hepatic functions in repeated alcohol consumption caused liver injuries. These findings are fully consistent with the histopathological findings in these 8 groups that received Asta-lipo. The serum biochemistry data suggested that p.o is an appropriate administration route for Asta-lipo to cure alcohol-induced liver disease, whereas i.p. is a more effective administration route for Asta-lipo to reverse liver injuries caused by repeated alcohol intake. Moreover, AST levels significantly dropped in the groups with oral and intraperitoneal Asta-lipo administration. More precisely, except in Groups H and I, AST levels in the groups that received oral and intraperitoneal administration of Asta-lipo for 2 and 4 weeks reduced to the normal range (Figure 8B).

Despite the decrease in the ALT levels in a few groups, intraperitoneal and oral administration of pure liposomes alone for 2 and 4 weeks could not cause ALT levels to return to the normal range (Figure 8C), suggesting that liposomes alone did not effectively exert reparative effects on alcohol-caused liver injuries. This is entirely coherent with histopathological observations in these groups. The AST levels in groups treated with liposomes alone through intraperitoneal and oral routes (Groups K~R, Figure 8D) showed somewhat notable results despite no significant differences in terms of AST levels. In the groups that received interspersed treatment with 30% alcohol prior to the 3-week treatment with 30% alcohol, administration with liposomes alone for 2 weeks through the intraperitoneal route (Group P) seemed to cause less deterioration and have some reparative effects on alcohol-induced liver injuries than that received 2-week liposomes alone via the oral route (Group O), which were also consistent with histopathological results. On the other hand, in the groups treated with 3-week 30% alcohol without repeated 30% alcohol, oral administration of pure liposomes (Groups K and M) caused lower ALT and AST levels compared to those administered via i.p. route (Groups L and N). In these situations, direct injection of lipids into abdominal cavity may lead to larger metabolic burden than oral intake of lipids since the lipids can be somewhat digested via oral intake. However, the mice liver in Group O should be quite largely degenerated or even damage after receiving repeated alcohol doses in a shorter duration (2 week), so extra liposomes/lipids intake orally could more easily increase hepatic index than those injected via i.p. Also, it was found that after 2-week and 4-week administration with Asta-lipo via p.o or i.p., the AST:ALT ratio was larger than 2 whereas the ratio in the mice received pure liposomes was still lower than 2 (Figure 8E,F).

## 3. Discussion

In the present study, the preventive and reparative effects of Asta-lipo and possible influences of liposomes on hepatic repair were examined and verified in vitro and in vivo using MTT assay, cell cycle analysis, histopathological and serum biochemical analyses. No significant influences were found in the cell cycle phases prior to treatment with pure liposomes or Asta-lipo. The in vitro results on the effects of Asta-lipo on the cell cycle of normal cells were in accordance with the data of the MTT assay despite slight decrease in L929 viability resulted from treatment with 0.01 mg/mL pure liposomes. These results are in line with our previous in vitro findings [12]. The investigation also showed that significant IC_50_ values below 0.5 mM were detected in vitro for phosphatidylcholines (PCs) like PC(12:0/12:0) and PC(14:1/14:1)trans [15]. Furthermore, the slight but significant inhibitory effects of pure liposomes on normal fibroblast viability could be because their inhibitory effects are non-cell cycle–specific instead of cell cycle–specific. These are in accordance with the minimal deteriorated influences of Asta-lipo on normal cells in the current study.

The body weight data of pure liposome-treated mice suggested that pure liposomes alone are not capable of exerting protective or therapeutic effects against repeated alcohol consumption, although liposomes themselves are not harmful to mice. The effects of Asta-lipo and pure liposomes in the mice prior to 3-week 30% alcohol with or without feeding with second dose of 30% alcohol were grossly and histopathologically analyzed and evaluated. These gross pathological observations primarily showed that Asta-lipo exerted ameliorative effects on alcoholic liver fibrosis in vivo. In the mice given treatment prior to 3-weeks of 30% alcohol consumption, our findings indicated a more complete restoration of hepatic tissues exerted by the 4-week treatment with intraperitoneal Asta-lipo (Figure 6F), but overall, this effect was weaker than that observed in mice treated with Asta-lipo orally for the same duration. Moreover, Masson’s Trichrome staining and semi-quantitation of collagen content also showed the lowest collagen level in the group orally administered Asta-lipo for 4 weeks (Group E, Figure 7U). Therefore, our histopathological findings in the mice prior to 3-weeks of 30% alcohol consumption suggested that oral administration of Asta-lipo for 2 and 4 weeks had reparative effects against alcoholic liver disease prior to alcohol consumption for 3 weeks.

On the other hand, the H&E findings in Groups C and H were in accordance with their ALT levels and indicated that their ALT levels returned within the normal range. However, the ALT levels in Group G significantly decreased but were still higher than the normal range compared with that in the group without any treatment; this result is coherent with the histopathological findings. The histopathological observations in the groups that received the 3-week alcohol treatment plus second-dose alcohol intake with intraperitoneal injection of Asta-lipo for 2 and 4 weeks (Groups H and J) revealed an obvious reduction in fibrotic areas, improvement in hepatocyte swelling, the rearrangement of hepatocytes, and the appearance of normal hepatic structures, such as the central vein and sinusoids, demonstrating that liver repair and recovery from the existing alcoholic liver disease plus repeated alcohol consumption were promoted and accelerated by intraperitoneal treatment with Asta-lipo (Groups H and J). These features were also found in the group that received second-dose alcohol and oral treatment with Asta-lipo for 4 weeks (Group I). Consistently, the ALT levels of Groups H, I, and J, drastically dropped to levels lower than their normal range, as collagen content of these groups were also clearly decreased (Figure 7U), which strongly evidenced that intraperitoneal treatment with Asta-lipo is capable of restoring liver injury and fibrosis in alcoholic liver disease with subsequent alcohol consumption, and this therapeutic effect can be active even in a relatively short period of 2 weeks.

In addition to inflammation and progressive fibrosis, hepatic angiogenesis has been found in chronic liver fibrosis regardless of its etiology. Evidence has shown that long-term ethanol intake is associated with angiogenesis through corresponding actions of several mediators, such as vascular endothelial growth factor (VEGF) and Vascular endothelial growth factor receptor 2 (VEGFR-2), in the liver [16,17]. In the group that received oral administration of pure liposomes for 2 weeks, obvious angiogenesis; inflammatory cell infiltration; and hepatocyte swelling, degeneration, and necrosis were observed in the mouse liver (Group K) (Figure 6K). Moreover, the liver of mice treated with intraperitoneal injection of pure liposomes for 2 weeks exhibited severe hepatocyte degeneration and necrosis as well as obvious hepatic angiogenesis, inflammatory cell infiltration, and fatty degeneration-like areas (Group L). Histopathological findings in Groups K and L indicated that treatment with pure liposomes, either through oral administration or intraperitoneal injection, could not restore liver injury and fibrosis in alcoholic liver disease. Instead, microscopic observations revealed that additional phospholipids from liposomal administration appeared to deteriorate liver injury and promote angiogenesis in existing alcoholic liver disease. With the continuation of liposomal treatment to week 4, the appearance of fibrotic tissues, inflammatory cell infiltration, hepatocyte degeneration and necrosis, and angiogenesis to certain extent remained in the liver (Groups M and N) (Figure 6M,N), showing that the administration of pure liposomes for a longer period could not exert any reparative effects on alcoholic liver disease. The ALT levels in these groups decreased but were still higher than the normal range, indicating the persistence of injury and loss of function in the liver. In addition, the livers of mice that received a repeated intake of 30% ethanol with oral administration of pure liposomes (Group O) for 2 weeks exhibited large areas of fatty liver change coupled with clear and possibly more acute hepatocyte swelling. In the counterpart as well as in the second-dose 30% ethanol intake group treated intraperitoneally with pure liposomes for 2 weeks (Group P), the livers presented with complex lesions of hepatocyte swelling, degeneration, and necrosis together with rather fibrotic tissues and inflammatory cell infiltration, implicating somewhat hepatic damage caused by intraperitoneally injected liposomes for a short period of 2 weeks. Group O showed an ALT level nearly equal to that of the nontherapy group, implicating acute toxicity mainly due to the oral intake of second-dose ethanol and to some degree through oral intake of pure liposomes. Oral administration with both treatments may more severely affect liver function or lead to larger burden for lipid metabolism and alcohol detoxification via portal circulation in a shorter period of 2 weeks in Group O than in Group P (i.p. treatment with pure liposomes). As pure liposomes were administered for 4 weeks (Group Q and R) through both oral and intraperitoneal routes in the experimental groups treated with 3-week alcohol intake plus a second dose of alcohol intake, severe hepatocyte necrosis and chronic hepatic fibrosis was noted, further suggesting that not only the loading of external lipids is ineffective in recovery from alcoholic liver disease caused by repeated alcohol consumption but also the extra lipids may worsen the conditions or hamper the natural reparative progress in the liver. This is fully supported by the finding that the ALT levels in Groups Q and R reached as high as that of the group without any treatment. The results obtained from Groups O to R revealed that treatment with pure liposomes is not only incapable of repairing existing alcohol-induced liver injury with or without ongoing ethanol intake but also that lipid-based liposomes cause more severe fatty degeneration and hepatocyte degeneration and necrosis, probably resulting from more oxidative stress and liver dysfunction [18,19]. (Figure 6O–R). By contrast, treatment with Asta-lipo through either administration routes for 4 weeks can obviously restore and attenuate hepatic necrosis, fibrosis, and fatty changes caused by 3-week alcohol intake, whereas oral administration of Asta-lipo exerted more rapid and efficient effects on hepatic repair, probably because oral administration enables Asta-lipo to directly undergo first-pass metabolism in the liver.

The ALT and AST data combined with in vivo histopathological findings also implicated that shorter-term 2-week administration of Asta-lipo via either administration routes was not sufficiently effective to steadily restore normal status from repeated alcohol consumption-induced hepatic damage. Based on the histopathological observations and serum ALT results, 4-week intraperitoneal or oral administration of Asta-lipo could effectively improve hepatic function and repair injured liver tissues caused by incurred and repeated consumption of alcohol. This could be because Asta-lipo administered intraperitoneally can distribute faster in the mice abdominal cavity, and the rescue of more severe, repeated alcohol-induced liver injuries can be more effectively attained. By contrast, oral administration of Asta-lipo may be capable of providing more steady and gradual liver function-recovering effects, which is more suitable for the treatment of mice that received 3-week 30% alcohol without repeated 30% alcohol intake. Moreover, recent investigations using animal models have shown different serum biochemistry data to the findings in humans [20]. Feeding male C57BL/6 mice with a Lieber-DeCarli diet containing 5% ethanol for 10 days, followed by a single dose of ethanol (5 g/kg body weight) by gavage, induces significant fatty liver and liver injury with peak serum levels of approximately 250 IU/L alanine aminotransferase and 420 IU/L aspartate aminotransferase 9 hours after gavage. Meanwhile, a second dose of 20% ethanol clearly led to increased ALT and AST [21]. Therefore, it is not surprising that in our study the mice that received 30% ethanol orally for 21 days showed ALT around 150 IU/L coupled with AST about 250 IU/L. Also, previous studies in humans have shown that the ratio of aspartate aminotransferase to alanine aminotransferase <1 suggest nonalcoholic steatohepatitis (NASH), a ratio of ≥2 is strongly suggestive of alcoholic liver disease [22]. In the mice p.o. or i.p. administered with Asta-lipo prior to 3-week 30% alcohol with or without feeding with second dose of 30% alcohol presented with AST: ALT values principally higher than 2, which is coherent with the previous conclusions by Sorbi et al. (1999) [22]. As to those mice administered with pure liposomes, Groups K, O, P and R showed the AST:ALT ratio less than 1, suggesting that subsequent administration with liposomes could be metabolic burdens under the liver conditions of 3-week 30% alcohol with or without feeding with second dose of 30% alcohol, leading to the AST:ALT ratio closer to NASH. However, longer-term (4 week) administration with liposomes prior to 3-week 30% alcohol with or without feeding with second dose of 30% alcohol, like the values shown in Groups M, N and Q, resulted in increased ratio of AST: ALT. This could be associate with the longer halflife of mitochondrial AST released in response to alcohol and the coexistence of deficiency of pyridoxal-6-phosphate in alcoholics, which is a cofactor for the enzymatic activity of ALT [23]. In addition, involvement of the non-invasive fibrosis markers for evaluation of ALD such as hyaluronan (HA) and Terminal peptide of procollagen III (PCIIINP) has been reported [24]. Although several non-invasive fibrosis markers have been suggested as alternatives to liver biopsy in patients with ALD, none has been sufficiently ascertained [25]. Despite the limitation of AST/ALT ratio in ALD patients together with severe fibrosis, animal studies showed different serum biochemistry data to these previous findings in humans [20] and our serum AST and ALT results showed the coherence to some extent with those found previously in mice [21]. The current study has extensively investigated and evaluated the therapeutic activities of Asta-lipo on the gross, histopathologic, and serum biochemical changes. The histopathologic observations and serum biochemical changes after Asta-lipo administration were evident and convincing in ALD on the mice model.

Asta has been shown to possess anti-inflammatory, free radical-clearing, and angiogenesis-inducing effects [26,27,28]. The liver is an organ with abundant blood supply and is in charge of clearing harmful substances in the circulatory system. Therefore, it is reasonable that previous studies have found that Asta has protective and therapeutic capabilities against liver fibrosis, steatohepatitis, and nonalcoholic fatty liver disease on murine models [8,29,30]. These actions have been found to be related with some growth factors such as TGF-β1, autophagy, and inhibition of the Smad3 pathway activation in hepatic stellate cells [31,32]. Furthermore, Asta has also been shown to protect liver injury induced by CCl_4_ in rats by suppressing oxidative stress parameters like malondialdehyde (MDA) and nitric oxide (NO), and enhancing the activities of physiological antioxidant system such as superoxide dismutase (SOD) and catalase (CAT) [33]. These protective effects of Asta against CCl_4_-induced liver damage are similar with those exerted by angiotensin-converting enzyme inhibitors [34]. To date, few studies have investigated the preventive and therapeutic effects of Asta on alcoholic liver diseases. The current data successfully highlighted that Asta-lipo administered intraperitoneally and orally possesses the capability to restore alcohol-induced liver injuries and has preventive actions on the repeated alcohol consumption status in vivo. We previously found that 2-week intraperitoneal administration of HA nanoparticles aggregated Asta was capable of restoring the retrorsine-CCl_4_-induced liver fibrosis and necrosis in a murine model, demonstrating that nanoparticle-encapsulated Asta can repair fibrotic and necrotic liver disease caused by toxicants in vivo [35]. Notably, in our previous study and in the current study, shorter-term (2 weeks) treatment with nanoparticle-encapsulated Asta through the intraperitoneal route exerted reparative effects on alcohol- or toxicant-induced liver injuries, indicating that the low bioavailability of Asta can be improved after nanoparticle encapsulation, and intraperitoneal is a feasible administration route for nanoparticle-encapsulated Asta to exhibit ameliorating actions against liver fibrotic and necrotic diseases.

## 4. Materials and Methods

### 4.1. Materials and Animals

Asta was produced and provided by Bioptik Technology Inc (Miaoli, Taiwan). Asta was extracted according to the method described in the US patent US8030523B2, filed by Bioptik Technology Inc. In brief, *Pomacae canaliculata* eggs were mixed proportionally in deionized water and homogenized (Polytron PT-2100, Bestway, Taiwan). The egg shells were then removed to obtain a solution of glycoprotein carotenoid. Proteins, sugars, and lipids in the glycoprotein carotenoid solution were sequentially removed to prepare the carotenoid solution. Finally, pure Asta was extracted using 95% ethanol, and the purity of Asta was quantified using the method described by Skrede et al. [36]. Cholesterol, lecithin, DSPC (molecular weight: 790.15 Da), chloroform, and methanol were obtained from Sigma (St. Louis, MO, USA). All chemicals used in this study were of reagent grade.

The animal research was approved by the Institutional Animal Care and Use Committee of I-Shou University, Taiwan (AUP-105-50-01), date (20-06-2016). A total of 90 wild-type C57BL/6J male mice (age, 12 weeks) were used in the mouse liver fibrosis model (average weight, 24 ± 2 g).

### 4.2. Production and Characterization of Astaxanthin-Encapsulated Liposomes

Asta-lipo was prepared using the evaporation sonication method with some modification [9]. Briefly, phospholipids used for liposomes were a mixture of DSPC (8.7 mg), cholesterol (2.9 mg), lecithin (30 μL), and Asta (0.5 mg). DSPC, cholesterol, lecithin, and Asta were dissolved in methanol:chloroform (1:1, *v/v*) and then loaded into a flask. The mixture in a flask was homogenized for 120 s (UP 200S, Germany) and dried using a rotary evaporator (N-1300, Japan) to form a thin layer of film. The produced film was rehydrated with deionized water (5 mL) and subjected to sonication for 20 min (Figure 1A). The size of Asta-lipo was calculated on a random sampling basis for approximately 100–150 individual liposomes through transmission electron microscopy (TEM). The entrapment efficiency (EE) of Asta in liposomes was determined as follows: 1 mL of the solution containing the prepared Asta liposomes was centrifuged at 14,000 rpm for 60 min, and the amount of nonencapsulated Asta in the supernatant was measured through high-performance liquid chromatography (HPLC, Agilent 1100 series). A standard concentration curve of Asta was established for determining the amount of Asta encapsulated in liposomes. The EE was calculated using the following formula:EE (%) = [(Total amount of Asta − Amount of nonencapsulated Asta)/Total amount of Asta)] × 100%

The in vitro release of Asta from liposomes was assessed. A total of 1 mL of Asta liposomes was added to a 1.5-mL microcentrifuge tube. The tube was then placed on a shaker with the temperature and shaking rate set at 37 °C and 40 rpm, respectively. At designated time points, the sample was centrifuged at 14,000 rpm for 60 min. The amount of nonencapsulated Asta in the supernatant was determined through HPLC. The independent experiment was repeated three times (*n* = 3). The in vitro release rate was calculated as follows:In vitro release (%) = [(Total amount of Asta − Residue of Asta)/Total amount of Asta] × 100%

### 4.3. In Vitro Cell Viability Study

L929 fibroblasts were suspended at a density of 1 × 10^4^ cells/mL and seeded onto 96-well plates. Asta-lipo, pure liposomes or control was added to each well in triplicate. The cell viability of L929 normal fibroblasts was examined using the 3-(4,5-Dimethylthiazol-2-yl)-2,5-diphenyltetrazolium bromide (MTT) assay. After a 24-h treatment, 20 μL of MTT solution was added to cells, and cells were then incubated for 3 h. The formazan precipitate was dissolved in 200 μL dimethyl sulfoxide (DMSO), and absorbance was measured at 570 nm on a multiplate reader (Thermo Scientific, Waltham, MA, USA).

### 4.4. Cell Cycle Analysis

The cell cycle of L929 fibroblasts was analyzed through flow cytometry after 24- and 48-h Asta-lipo treatment. L929 fibroblasts (1.5 × 10^5^ cells/mL) were seeded onto the flask for 24 h; then, 500 μL of Asta-lipo or pure liposomes was added. The cells were incubated (37 °C, 5% CO_2_) for 24 or 48 h before harvest. Thereafter, the cells were centrifuged at 1200 rpm for 5 min and washed with phosphate-buffered saline, followed by staining with PI/RNase staining buffer (BD Bioscience, San Jose, CA, USA) for 30 min at room temperature. The cell cycle was analyzed through flow cytometry (Accuri C6, BD Bioscience). The cell cycle distribution was measured from 10,000 cells by using the ModFit LT software (Topsham, ME, USA).

### 4.5. Animal Experimental Design

Mice received an intragastric dose of 30% ethanol (*v*/*v*) at a dose of 3 g/kg body weight every day for 21 days through a 24-gauge stainless feeding tube. Mice were weighed on alternate days and housed under standard conditions, with food and water provided ad libitum. Mice with alcohol-induced liver fibrosis were categorized into different experimental groups according to their treatment period, treatment type (Asta-lipo or pure liposomes), and administration route, as shown in Table 2. Intraperitoneal injection or oral administration of Asta-lipo/pure liposomes were executed as shown in Figure 1B. Before intraperitoneal injection of treatments, mice were anaesthetized with Zoletil^®^ (tiletamine with zolazepam, 40 mg/kg, i.p.) and xylazine (10 mg/kg, i.p.). At the end of experiments, the weight of mice in each group was measured; then, mice were euthanized through an overdose of CO_2_. Blood and liver samples were harvested for serum biochemical and histopathological analyses, respectively. Figure 1C showed schematic representation and time scale of the establishment of alcoholic disease and administration routes on the mouse model. 

### 4.6. Histopathological Analysis

After a 2-week and 4-week treatment with Asta-lipo or pure liposomes, the livers of mice in the respective groups were collected. The liver lobes were removed, and liver tissues were fixed in 10% neutral buffered formalin. The liver cuts were dehydrated in ethanol and then embedded in wax. Hematoxylin and Eosin (H&E) staining was performed in dishes, starting with a dewaxing procedure in xylene. Sequential incubations in 100% and 95% ethanol and tap water were conducted to rehydrate the liver tissues. The liver tissues were stained with Mayer’s hematoxylin (Merck, Darmstadt, Germany) for 15 min, followed by a 3-min wash in tap water. The tissues were then stained with eosin–phloxine solution (Merck) for 30 s and washed in tap water for 30 min. The liver tissue samples were then dehydrated with sequential incubations in 50%, 70%, 80%, 90%, 95%, and 100% ethanol and xylene.

Histological sections were further stained with Masson’s Trichrome to assess collagen content in mice liver after induction of alcoholic liver disease with or without treatment with Asta-lipo or pure liposomes. The liver tissue sections were deparaffinized in xylene and then hydrated in graded ethanol, xylene, and distilled water. Weigert’s iron hematoxylin solution was used to stain the liver tissues. Tissue slides were then sequentially incubated in Biebrich scarlet-acid fuchsin (Sigma) for 5 min, followed by treatment with phosphomolybdic–phosphotungstic acid solution and aniline blue (Sigma). ImageJ software was applied to measure the collagen content in each group [37]. In brief, liver samples from experimental groups were stained and compared to Group B, S or T. The color settings in the ImageJ software was maintained at all times between the calculations of the blue-stained areas in samples. Magnification 100× was employed to evaluate the samples and the calculation was repeated in four microscopic fields.

### 4.7. Measurement of Alanine Aminotransferase and Aspartate Aminotransferase

We executed the ALT and AST assays and calculate the ratio of AST:ALT to further clinically evaluate and validate the status or recovery from alcohol-induced liver fibrosis and damage after treatment with Asta-lipo or pure liposomes in the experimental groups. ALT or AST samples were prepared according to the instruction protocol (AAT Bioquest, CA, USA). Each 100 μL mixture of the sample and ALT/AST enzyme on the plate were incubated at 37 °C for 20–30 min in the dark. The absorbance was measured at 575 nm on a microplate reader.

### 4.8. Statistical Analysis

Data are presented as the mean ± standard error of the mean (SEM) unless stated otherwise. Results were analyzed using one-way analysis of variance (ANOVA) followed by the Tukey–Kramer multiple comparisons test on SPSS version 17.0 (SPSS Inc., Chicago, IL, USA) to determine whether significant differences existed between the control and experimental groups (*P* < 0.05) [38].

## 5. Conclusions

The current study evidenced that Asta-lipo exerts protective and restorative effects against existing alcohol-induced liver injury with or without ongoing alcohol consumption, suggesting that liposomal encapsulation is an applicable modification in the drug delivery system for Asta and that Asta-lipo may be a promising therapeutic agent for alcohol consumption-induced liver diseases. In addition to the free radical-scavenging, antitumor, anti-inflammatory, and NASH-therapeutic capabilities of Asta, the convincing data in our current work successfully verifies that Asta-lipo exerts reparative and protective effects against alcoholic liver injuries and may be a potential therapy for alcoholic liver disease.

## Figures and Tables

**Figure 1 ijms-20-04057-f001:**
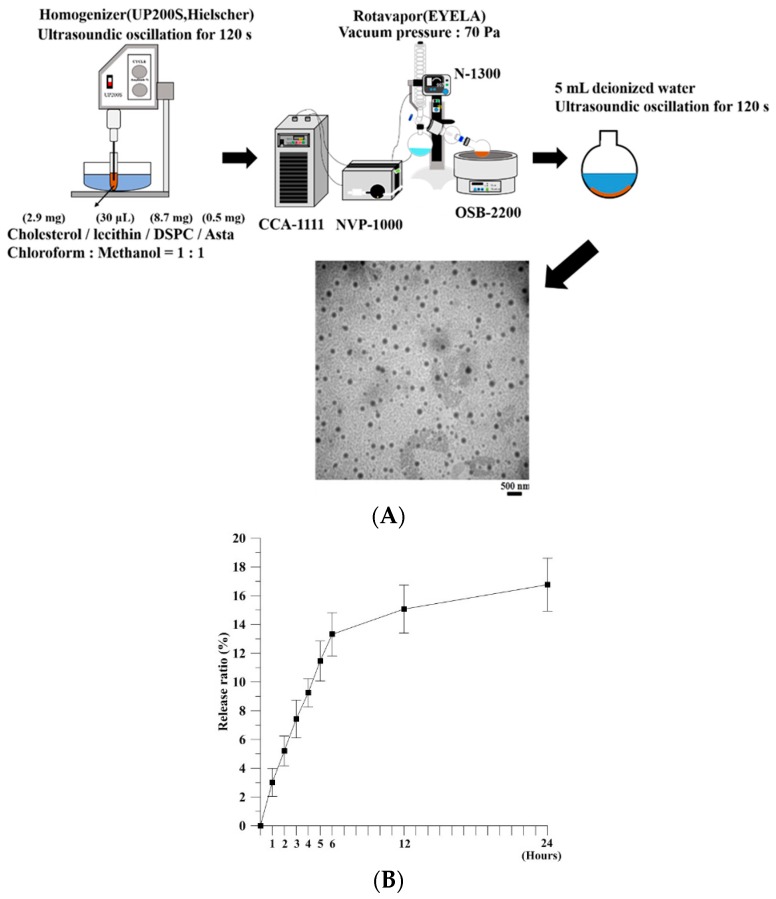
(**A**) Schematic of the Asta-lipo fabrication processes and the TEM images of the produced Asta-lipo. The TEM shows that the diameter of the fabricated Asta-lipo was approximately 225.0 ± 58.2 nm (mean ± SD). (**B**) Profile of Asta released from Asta-lipo (*N* = 3, data are presented as the mean ± SEM by one-way analysis of variance (ANOVA)). (**C**) Schematic representation and time scale of the establishment of alcoholic disease and administration routes on the mouse model (scale bar: day).

**Figure 2 ijms-20-04057-f002:**
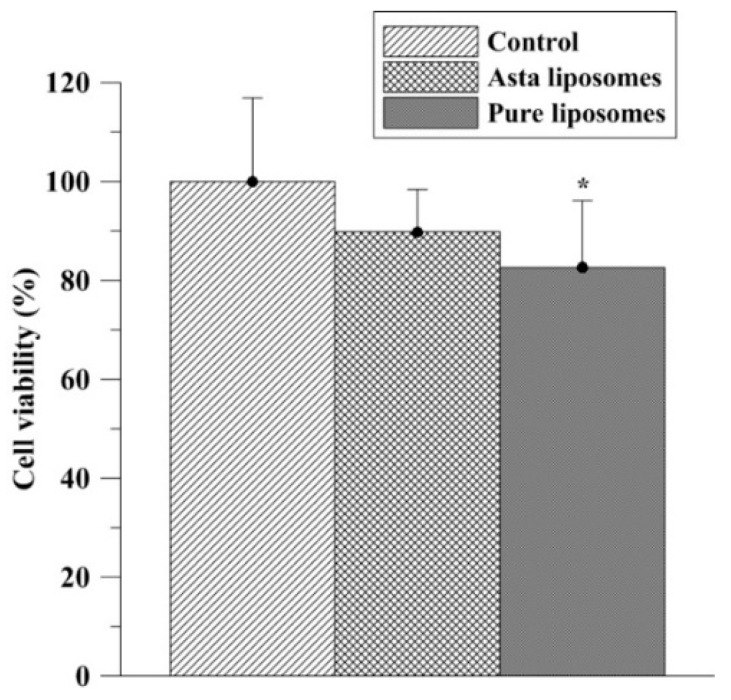
Effects of pure liposomes and Asta liposomes (Asta-lipo) at 0.01 mg/mL on the viability of L929 fibroblasts assessed using the 3-(4,5-dimethylthiazol-2-yl)-2,5-diphenyltetrazolium bromide (MTT) assay after 24-h incubation. Control stands for incubation with the vehicle for the same period (24 h) as Asta-lipo and pure liposomes. Pure liposomes slightly but significantly reduced the viability of the normal fibroblasts (*N* = 3, data are presented as the mean ± SEM, * *P* < 0.05 by ANOVA compared to control).

**Figure 3 ijms-20-04057-f003:**
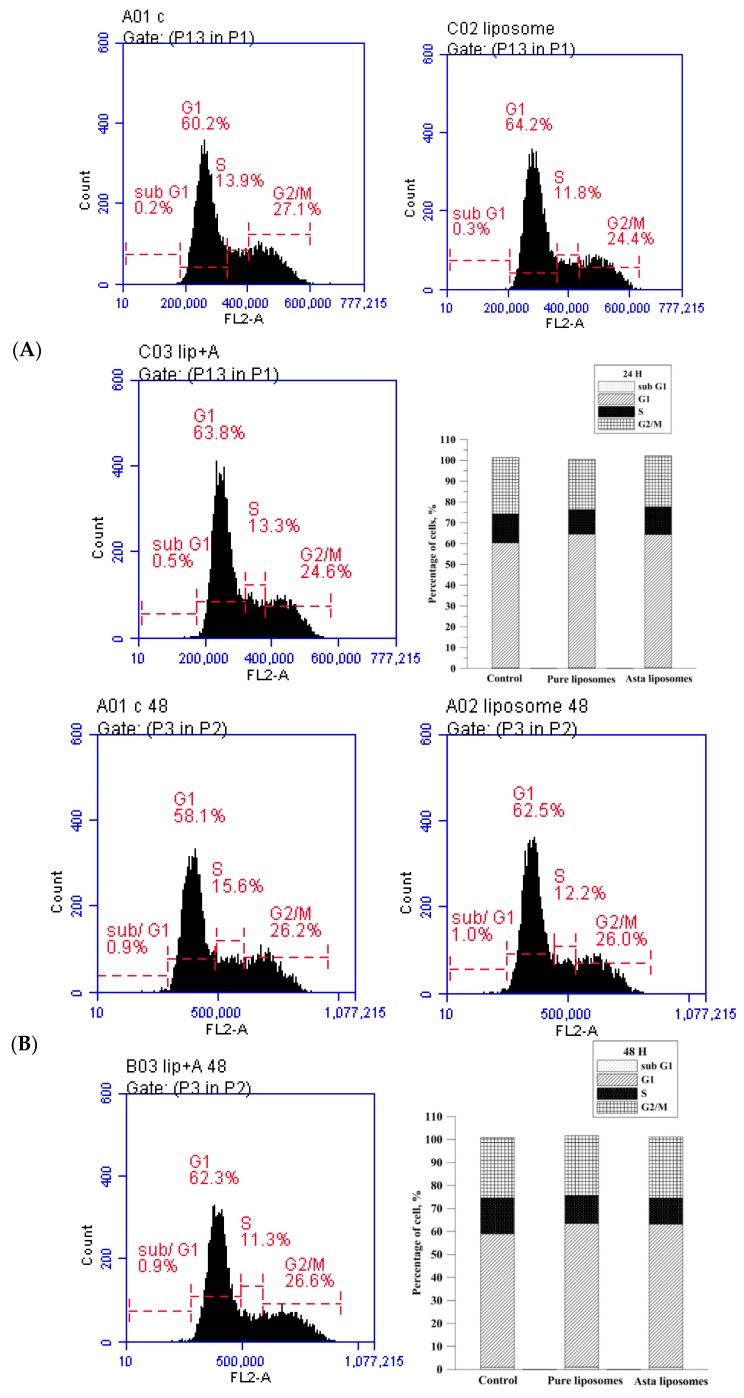
Cell cycle distribution analysis of L929 fibroblasts after (**A**) 24- and (**B**) 48-h treatment with 0.01 mg/mL pure liposomes or Asta-lipo. No significant differences were found in the cell cycle phases prior to treatment with pure liposomes or Asta-lipo.

**Figure 4 ijms-20-04057-f004:**
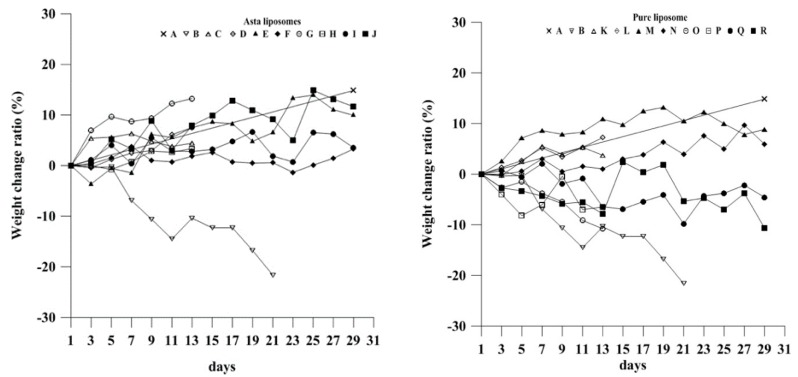
The weight change ratio of mice in each experimental group. Mice that received Asta-lipo gained weight normally, whereas mice in Groups O–R that received repeated alcohol intake lost weight, which was similar to the result in the untreated mice in Group B (Groups which received 2-week p.o and i.p. administration with Asta-lipo or pure liposomes were compared to Group S; Groups which received 4-week p.o and i.p. administration with Asta-lipo or pure liposomes were compared to Group T; *N* = 4, * *P* < 0.05, ** *P* < 0.01 and *** *P* < 0.001, by ANOVA).

**Figure 5 ijms-20-04057-f005:**
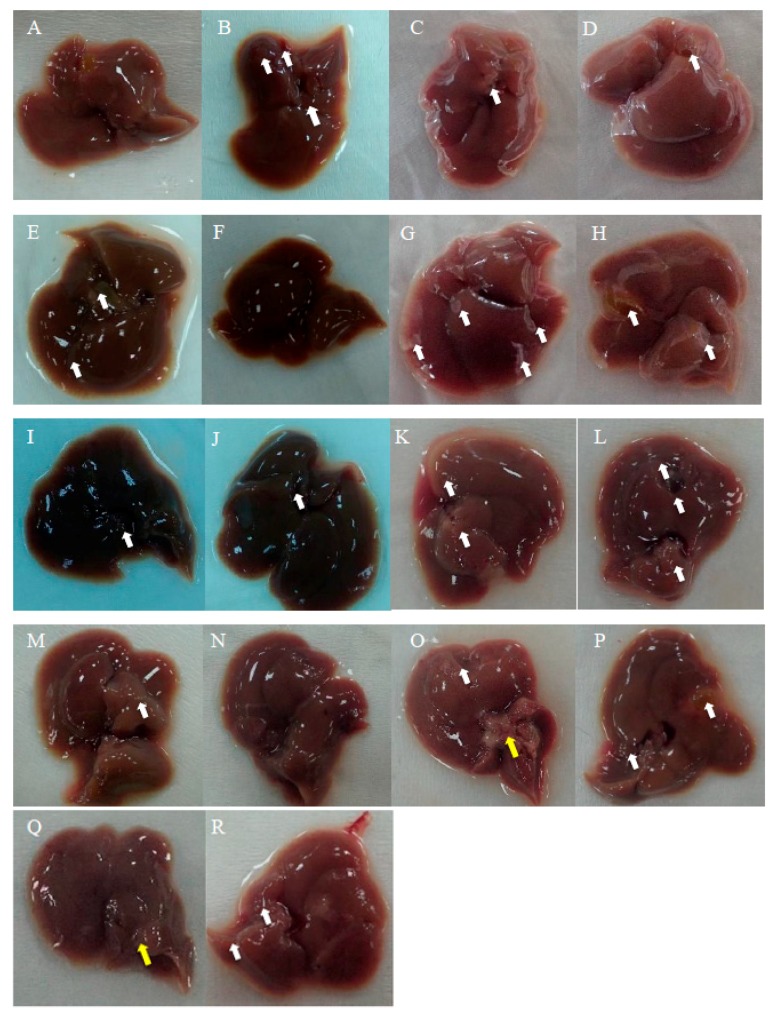
Gross observations of the liver in mice with or without treatment with Asta-lipo or pure liposomes for 2 and 4 weeks on the mouse model that received two modes of alcohol administration. The categorization of the experimental groups is shown in Table 1. Group **A** shows the liver of normal mice, and Group **B** received alcohol for 3 weeks to induce alcoholic disease. Mice were given Asta-lipo or pure liposomes for 2 or 4 weeks prior to the administration of 30% alcohol for 3 weeks (Groups **C**–**F** and **K**–**N**) or 3-week alcohol administration plus a second dose of 30% alcohol (Groups **G**–**J** and **O**–**R**). White arrows indicate the enlarged liver lobe coupled with some white deposits, and yellow arrows show the locations of fibrotic tissues.

**Figure 6 ijms-20-04057-f006:**
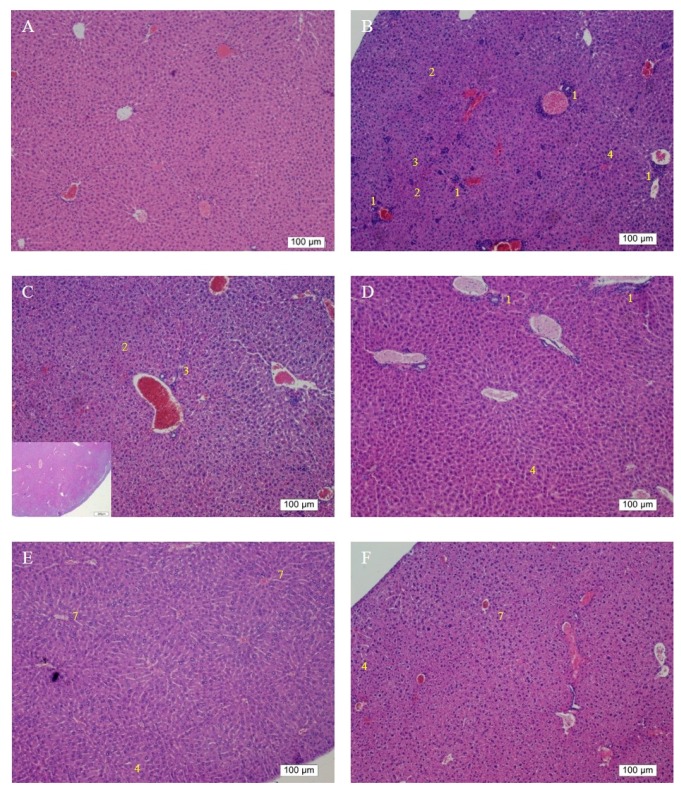
H&E staining and histopathological analysis of the liver in the experimental mice that received alcohol through two modes. Group **A** showed normal mice liver as Group **B** represented the mice received 3-week 30% alcohol feeding. Mice were given Asta-lipo or pure liposomes for 2 or 4 weeks prior to the 3-week 30% alcohol consumption (Groups **C**–**F** and **K**–**N**) or 3-week 30% alcohol administration plus a second dose of 30% alcohol (Groups **G**–**J** and **O**–**R**). Mice in Groups S and T received alcohol for 3 weeks without any administration for 2 and 4 weeks, respectively. Histopathological lesions were observed at 100× magnification as below: 1. inflammatory cell infiltration. 2. fibrotic tissues/accumulation of fibrocytes. 3. necrosis of hepatocytes. 4. swollen hepatocytes. 5. fatty degeneration-like areas. 6. proliferation of minibile duct. 7. hepatic repair showing normal hepatic structures such as the central vein, hepatocytes, and sinusoids (*N* = 4).

**Figure 7 ijms-20-04057-f007:**
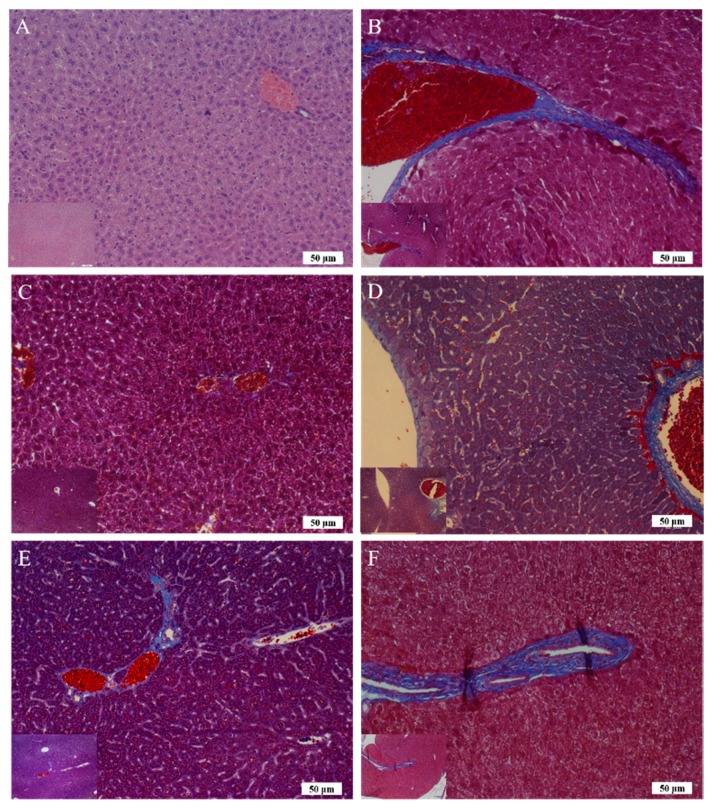
(**A**–**T**) Masson’s Trichrome staining of the liver sections in the experimental mice that received alcohol through two modes. The observations were performed at 200× magnification (the image on bottom left site was 100× magnification). Group **A** showed normal mice liver as Group **B** represented the mice received 3-week 30% alcohol feeding. Mice were given Asta-lipo or pure liposomes for 2 or 4 weeks prior to the 3-week 30% alcohol consumption (Groups **C**–**F** and **K**–**N**) or 3-week 30% alcohol administration plus a second dose of 30% alcohol (Groups **G**–**J** and **O**–**R**) as shown in Table 1. (**U** and **V**). Mice were treated with Asta-lipo (subfigure **U**) or pure liposomes (subfigure **V**) for 2 and 4 weeks and the collagen content in liver of each group was semi-quantitated by ImageJ software (Version 1.50, National Institutes of Health, USA). The color settings in ImageJ software was maintained at all times between the calculations of the blue-stained areas in samples. Magnification 100× was employed to evaluate the samples and the calculation was repeated in four microscopic fields. The measurement was repeated in four microscopic fields. (Group **A** and the treated groups were compared to Group **B**; Groups which received 2-week p.o and i.p. administration with Asta-lipo or pure liposomes were compared to Group S; Groups which received 4-week p.o and i.p. administration with Asta-lipo or pure liposomes were compared to Group T; *N* = 4, *,#,+ *P* < 0.05, ##,++ *P* < 0.01 and *** *P* < 0.001, by ANOVA).

**Figure 8 ijms-20-04057-f008:**
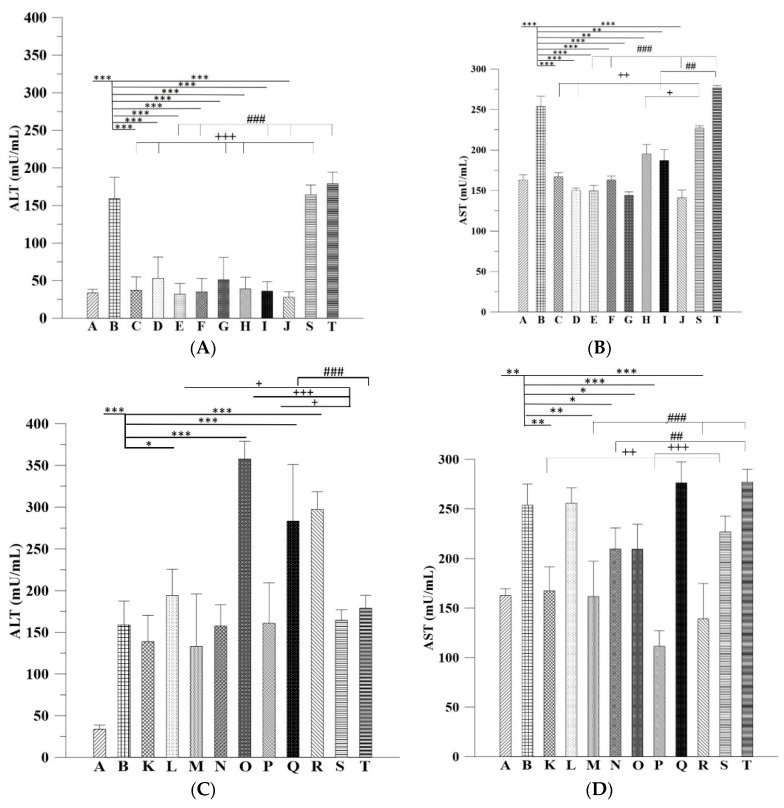
ALT, AST levels and the AST:ALT ratio of mice treated with Asta-lipo (subfigures **A**, **B** and **E**) or pure liposomes (subfigures **C**, **D** and **F**) for 2 and 4 weeks. Mice were given Asta-lipo or pure liposomes for 2 or 4 weeks prior to the 3-week 30% alcohol consumption (Groups **C**–**F** and **K**–**N**) or 3-week 30% alcohol administration plus a second dose of 30% alcohol (Groups **G**–**J** and **O**–**R**) as shown in Table 1. The letters A to T in each subfigure represent Groups **A** to Group **T** where the treatment conditions of each group are shown in Table 1 (Group **A** and the treated groups were compared to Group **B**; Groups which received 2-week p.o and i.p. administration with Asta-lipo or pure liposomes were compared to Group S; Groups which received 4-week p.o and i.p. administration with Asta-lipo or pure liposomes were compared to Group T; *N* = 4, *,#,+ *P* < 0.05, **,##,++ *P* < 0.01 and ***,###,+++ *P* < 0.001, by ANOVA).

**Table 1 ijms-20-04057-t001:** Analysis of cell cycle distribution of normal L929 fibroblasts exposed to Asta-lipo and pure liposomes for 24 and 48 h.

24 H	Sub G1	G1	S	G2/M
**Control**	0.2%	60.2%	13.9%	27.1%
**Asta-lipo**	0.5%	63.8%	13.3%	24.6%
**Pure liposomes**	0.3%	64.2%	11.8%	24.4%
**48 H**				
**Control**	0.9%	58.1%	15.6%	26.2%
**Asta-lipo**	0.9%	62.3%	11.3%	26.6%
**Pure liposomes**	1.0%	62.5%	12.2%	26.0%

**Table 2 ijms-20-04057-t002:** Experimental groups and their treatment conditions on the mouse model of alcoholic liver disease.

Group	Treatment
**A**	Normal liver
**B**	The mice received 3-week 30% alcohol feeding
**C**	Oral administration of Asta-lipo for 2 weeks (one treatment/per 2-d)
**D**	Intraperitoneal injection of Asta-lipo for 2 weeks (one treatment/per 2-d)
**E**	Oral administration of Asta-lipo s for 4 weeks (one treatment/per 2-d)
**F**	Intraperitoneal injection of Asta-lipo for 4 weeks (one treatment/per 2-d)
**G**	Oral administration of Asta-lipo for 2 weeks *
**H**	Intraperitoneal injection of Asta-lipo for 2 weeks *
**I**	Oral administration for of Asta-lipo 4 weeks *
**J**	Intraperitoneal injection of Asta-lipo for 4 weeks *
**K**	Oral administration of pure liposomes for 2 weeks (one treatment/per 2-d)
**L**	Intraperitoneal injection of pure liposomes for 2 weeks (one treatment/per 2-d)
**M**	Oral administration of pure liposomes for 4 weeks (one treatment/per 2-d)
**N**	Intraperitoneal injection of pure liposomes for 4 weeks (one treatment/per 2-d)
**O**	Oral administration of pure liposomes for 2 weeks *
**P**	Intraperitoneal injection of pure liposomes for 2 weeks *
**Q**	Oral administration for of pure liposomes for 4 weeks *
**R**	Intraperitoneal injection of pure liposomes for 4 weeks *
**S**	The mice received 3-week alcohol feeding but without treatment for 2 weeks
**T**	The mice received 3-week alcohol feeding but without treatment for 4 weeks

* Cycles of 2 days: treatment with Asta-lipo or pure liposomes on day 1, followed by feeding of 30% ethanol on day 2.

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
