# Peer review of "Therapeutic and Protective Effects of Liposomal Encapsulation of Astaxanthin in Mice with Alcoholic Liver Fibrosis"

_ijms, 2019, doi:10.3390/ijms20164057_

Round 1

Reviewer 1 Report

Comments:

This manuscript, ID ijms-542710, titled “Therapeutic and Protective Effects of Liposomal Encapsulation of Astaxanthin in Mice with Alcoholic Liver Fibrosis" for the International Journal of Molecular Sciences. It presents a study, the marine carotenoid astaxanthin is incorporated into liposomes to improve its stability and allow delivery to the liver to treat alcoholic liver fibrosis in mice.

 In this study the research strategy was appropriate, with the methods clearly described, the results clearly presented and discussed, properly supporting the conclusions reached. This work should be fit for publication after the following minor revisions:

1.   Page 2, Line 57: The first sentence of this paragraphs needs a reference.  

2.   Page 2, Line 71: “DSPC liposome-encapsulated propranolol was employed as a carrier” should be replaced with “DSPC-based liposomes were employed as carriers”.

3.   Figure 1B and Page 2, Line 84: The data points and error bars are not properly defined. Although a value of 225.0±58.2 nm is presented, these values are not defined, e.g., mean±SEM.

4.   Figure 2: The data points and error bars are not properly defined.

5.   Figure 2: The nature of the control should be defined.

6.   Figure 7: The data points and error bars are not properly defined. The number of measurements N is also not defined.

7.   Page 11, Line 217: The use of ALT as a biomarker for liver injury should be properly supported by a reference.

8.   Page 13, Line 242: It is mentioned that there was a “slight decrease in L929 viability resulted from treatment with 0.01 mg/mL pure liposomes,” when actually, this effect was significant. This significant difference should be properly addressed.

9.   Page 16, Line 402: Cell viability was tested using three formulations, asta-lipo, pure-lipo and control. Only the first is mentioned and the control is not defined.

10.Page 16, Line 418: The experimental design showing how liver fibrosis is induced in this animal model should be supported by a reference. Including an explanation on why this specific method was chosen for this study.

Author Response

Reviewer 1:

Comments:

This manuscript, ID ijms-542710, titled “Therapeutic and Protective Effects of Liposomal Encapsulation of Astaxanthin in Mice with Alcoholic Liver Fibrosis" for the International Journal of Molecular Sciences. It presents a study, the marine carotenoid astaxanthin is incorporated into liposomes to improve its stability and allow delivery to the liver to treat alcoholic liver fibrosis in mice.

In this study the research strategy was appropriate, with the methods clearly described, the results clearly presented and discussed, properly supporting the conclusions reached. This work should be fit for publication after the following minor revisions:

1.     Page 2, Line 57: The first sentence of this paragraphs needs a reference. 

Response: We greatly appreciate the reviewer’s comment.

An appropriate reference has been cited [9]- Akbarzadeh, A.; Rezaei-Sadabady, R.; Davaran, S.; Joo, S. W.; Zarghami, N.; Hanifehpour, Y.; Samiei, M.; Kouhi, M.; Nejati-Koshki, K., Liposome: classification, preparation, and applications. Nanoscale Res Lett 2013, 8, (1), 102.

2.     Page 2, Line 71: “DSPC liposome-encapsulated propranolol was employed as a carrier” should be replaced with “DSPC-based liposomes were employed as carriers”.

Response: We greatly appreciate the reviewer’s comment. The sentence has been revised.

3.     Figure 1B and Page 2, Line 84: The data points and error bars are not properly defined. Although a value of 225.0±58.2 nm is presented, these values are not defined, e.g., mean ± SEM.

Response: We greatly appreciate the reviewer’s comment. The regarding data points and error bars (mean ± SEM) have been defined in revised text and Fig. 1 legend.

4.     Figure 2: The data points and error bars are not properly defined.

Response: We greatly appreciate the reviewer’s comment. The regarding data points and error bars (mean ± SEM) have been defined in Figure. 2 legend.

5.     Figure 2: The nature of the control should be defined.

Response: We greatly appreciate the reviewer’s comment. The data points, typographical errors and the definition of control have been added to the Figure 2 legends.

6.     Figure 7: The data points and error bars are not properly defined. The number of measurements N is also not defined.

Response: We greatly appreciate the reviewer’s comment. The data points, error bars and number of N in Fig. 7 have been revised. Because we added a new figure in the revised manuscript, this figure 7 shifted to figure 8. All changes were corrected in the revised manuscript in red color.

7.     Page 11, Line 217: The use of ALT as a biomarker for liver injury should be properly supported by a reference.

Response: We greatly appreciate the reviewer’s comment. An appropriate reference has been cited [14]- Kim, W. R.; Flamm, S. L.; Di Bisceglie, A. M.; Bodenheimer, H. C.; Public Policy Committee of the American Association for the Study of Liver, D., Serum activity of alanine aminotransferase (ALT) as an indicator of health and disease. Hepatology 2008, 47, (4), 1363-70.

8.     Page 13, Line 242: It is mentioned that there was a “slight decrease in L929 viability resulted from treatment with 0.01 mg/mL pure liposomes,” when actually, this effect was significant. This significant difference should be properly addressed.

Response: We greatly appreciate the reviewer’s comment. The effects have been properly addressed: These results are coherent with our previous in vitro findings [12] while investigation also showed that significant IC50 values below 0.5 mM were detected in vitro for phosphatidylcholines (PCs) like PC (12:0/12:0) and PC(14:1/14:1) trans [15].

9.     Page 16, Line 402: Cell viability was tested using three formulations, asta-lipo, pure-lipo and control. Only the first is mentioned and the control is not defined.

Response: We greatly appreciate the reviewer’s comment. Three formulations/groups in the MTT assay have been mentioned and the control has been defined.

10.  Page 16, Line 418: The experimental design showing how liver fibrosis is induced in this animal model should be supported by a reference. Including an explanation on why this specific method was chosen for this study.

Response: We greatly appreciate the reviewer’s comment. The in vivo experimental design in the current study is for simulation of human alcoholic liver disease. The induction pattern of alcohol-induced liver injury in mice was similar to the protocols used in the previous report [1](Zhou et al., 2017) with some modifications (a slightly lower dose but a little longer induction period). Alcohol drinkers consuming up to 2 drinks/day (men) or 1 drink/day (women) are defined as moderate drinkers and do not disclose increased risk of organ damage compared to abstainers. Daily consumption above those limits can lead to health, personal and social problems (Mathurin and Bataller, 2013). Moderate- to long-term heavy alcoholic consumption and recurrence of alcoholic intake in the patients with heavy alcoholic addiction should be extremely noted since alcoholic liver disease (ALD) is the leading cause of death among adults with excessive alcohol consumption [2] (Mathurin and Bataller, 2013).

1. Zhou, T.; Zhang, Y. J.; Xu, D. P.; Wang, F.; Zhou, Y.; Zheng, J.; Li, Y.; Zhang, J. J.; Li, H. B., Protective Effects of Lemon Juice on Alcohol-Induced Liver Injury in Mice. Biomed Res Int 2017, 2017, 7463571.

2. Mathurin, P.; Bataller, R., Trends in the management and burden of alcoholic liver disease. J Hepatol 2015, 62, (1 Suppl), S38-46.

Reviewer 2 Report

Comments to the authors

Wu and colleagues studied the protective effects of Asta-lipo on alcohol-induced liver fibrosis. As mentioned in the manuscript; “over the past decade, several studies have investigated the preventive or therapeutic effects of Asta against various disorders or diseases, such as cancer, age-related macular degeneration, inflammation, and cardiovascular oxidative stress, as well as have examined its effects for the promotion of immune responses. Asta has also been found to be capable of preventing diet-induced obesity and hepatic steatosis in vivo, more evidenced by it's stronger preventive and nonalcoholic steatohepatitis (NASH) therapeutic effects than those exerted by vitamin E on a mouse model”. I found the protective effect of Asta in liver diseases is not so novel in the current study. However, the fabrication and use of liposome encapsulated Asta looks new and may be useful for the application in the drug delivery system which looks interesting.

The author needs to answer and correct the following concerns before accepting for the publication.

Major concerns

1. The authors tried to demonstrate the therapeutic effects of Asta-lipo on alcoholic liver fibrosis in mice. They have shown the effect of Asta-lipo to recover the hepatic fibrosis induced by alcohol consumption. However, I did not find any results indicating hepatic fibrosis caused by alcohol and restoration after Asta-lipo treatment. The gross observations of the liver are not sufficient to show the fibrosis and other injuries. The authors are suggested to perform some fibrosis-related protein expression (e.g., alpha-SMA or any other collagen types); and liver sections showing fibrosis by Masson’s trichrome or Sirius Red staining.

2. The authors demonstrated that the increased serum ALT and AST as an indication of alcohol-induced liver disease which was further decreased after the Asta-lipo treatment.

Ø  In reference to Kim et al., https://doi.org/10.1002/hep.22109, high ALT level is correlated with the severity of non-alcoholic fatty liver disease (NAFLD), not with alcoholic liver disease (ALD). This current study is focusing on ALD and its consequences on liver injury and fibrosis. How can an author discuss this point? The author should compare and discuss the ALD with NAFLD in detail at least in the discussion section explaining ALT and AST.

Ø  Additionally, the ratio of AST: ALT is recommended to measure in all groups.

Ø  Likewise, there are several non-invasive fibrosis markers as suggested by previous papers (DOI: 10.3748/wjg.v21.i39.11044; 10.1155/2014/357287). Why did the authors specifically perform only ALT and AST as a marker for liver injury and fibrosis in ALD? It needs to be clearly explained in the manuscript.

3. The manuscript definitely lacks the molecular mechanisms; how Asta-lipo reduces the liver fibrosis in mice having alcoholic liver disease. The serum ALT and AST are just indicators of liver injury and not possesses a therapeutic mechanism to describe it. Is it possible to show the molecular mechanism/cellular signaling to make the manuscript smoother and easier to understand for all the readers?

Ø  The lines 346-354 of discussion section explain some of the mechanisms for the protective effect of Asta in previous papers. The protective effects of Asta are already known by various mechanisms, however, in the current study, the authors have used Asta-lipo which is the main focus of the study. The protective mechanism of Asta-lipo is not explained in the discussion. So, the authors need to show some mechanisms by performing experiments to show the effect of Asta-lipo in alcohol-induced liver disease.

4. The explanation of Figure 7C and 7D is missing in the result section. It needs to be explained in the result section.

5. The model of group B (Fig 7C) is same as of group B (Fig 7A), but the data value looks varied, though they have measured ALT in both figures. Can the author explain about this point? Why there is variation in the error bars? The error bar of Fig 7A (group B) is around 190 mU/mL and the error bar of Fig 7C (group B) is around 175 mU/mL.

6. In the discussion section, the authors have explained more about their results and less about the previously published literature. The majority of the results should be explained in the result section than in the discussion. Several points are repeated in the discussion from result sections. The authors are suggested to discuss more on the literature published in these areas, reduce the content which has already explained in the result sections and add more references to support the study.

Other comments

1. What experiments were done for the ‘Serum biochemical data’ in Abstract portion Line 23? Need to explain the name of the experiments.

2. In the Introduction section Line 76-80, the sentence looks so long and difficult to catch the meaning. It’s better to split into two sentences.

3. Under animal experimental design (Line 418), the authors have explained the treated dose of ethanol in a microliter, which is very unusual. It should be in mL/kg or µL/kg or g/kg (b.w). See some of the examples in these papers https://doi.org/10.1155/2017/7463571; 10.1016/j.jhep.2016.11.004, https://doi.org/10.3390/ijms17101616.

4. Since the author mentioned about the statistical differences in body weight of different treatment groups (in result sections), the sign of statistical analysis is required in Figure 5 to show the differences among the groups and explanation in figure legends.

5. How Asta-lipo or only liposomes were dissolved? What was the solvent used for oral and IP injection? Did the authors use enough control mice for the comparisons among the groups? What is the meaning of Normal Liver in Group A? The mice of this group were untreated (normal) or were treated with the vehicle?

6. In Figure 7, the explanation of figure legend is not enough. The author should explain the figures in more detail. What does A, B, C, D,……T means? It needs explanation in the figure legend too.

7. The statistical analysis of Figure 7 looks very confusing. To make easier for the readers, the authors should clearly mark and explain where they like to show the comparisons between the groups by different signs. They need to explain in the figure legends about the compared groups. One asterisk is seen near 350 unit (looks very unusual) in Figure 7C. What does that mean? They need to mention the ‘n’ number of mice in the figure legends also. 

8. What does p.o stand for? The full name of p.o is missing. What is the difference between Group B (Without p.o or i.p administration) and Group S and T (Without treatment for 2 weeks, without treatment for 4 weeks)?

9. In Figure 6, the appearance of histological data is not clear. The magnification of the liver cells looks very small so that it makes difficulties to identify the inflammatory cells, fibrosis, necrosis or the repairment. Instead, high magnification images are suggested for the authors.

10. The author claims that the magnification of the images of Figure 6 is same i.e. 100x. However, the figure 6A shows different magnification image (50 µm) than other images (100 µm). The consistency is required throughout the figures and needs correction in the figure legend.

11. The schematic diagram of Figure 1C is not clear. Please add more description of the materials dose and name details on the figure as well as legends. In addition, if you show the H and E data on the alcohol-induced liver damage, then it is expected to show the image of the treated conditions.

12. The protocol for histopathological analysis is not fully explained. “The tissues were then stained with eosinphloxine solution (Merck) for 30 s” cannot be the final steps of this staining protocol. The letter of Eosin should be capital in Line 432.

13. In materials and methods section 4.7, the explanation of ALT and AST (from Line 438-446) is not suitable in this section. Better to keep in the introduction part.

14. Some spelling mistakes were found such as radical-savaging (Line 462). Please check the grammatical and typos error throughout the paper.

Author Response

Reviewer 2:

Comments to the authors

Wu and colleagues studied the protective effects of Asta-lipo on alcohol-induced liver fibrosis. As mentioned in the manuscript; “over the past decade, several studies have investigated the preventive or therapeutic effects of Asta against various disorders or diseases, such as cancer, age-related macular degeneration, inflammation, and cardiovascular oxidative stress, as well as have examined its effects for the promotion of immune responses. Asta has also been found to be capable of preventing diet-induced obesity and hepatic steatosis in vivo, more evidenced by it's stronger preventive and nonalcoholic steatohepatitis (NASH) therapeutic effects than those exerted by vitamin E on a mouse model”. I found the protective effect of Asta in liver diseases is not so novel in the current study. However, the fabrication and use of liposome encapsulated Asta looks new and may be useful for the application in the drug delivery system which looks interesting.

The author needs to answer and correct the following concerns before accepting for the publication.

Major concerns

1. The authors tried to demonstrate the therapeutic effects of Asta-lipo on alcoholic liver fibrosis in mice. They have shown the effect of Asta-lipo to recover the hepatic fibrosis induced by alcohol consumption. However, I did not find any results indicating hepatic fibrosis caused by alcohol and restoration after Asta-lipo treatment. The gross observations of the liver are not sufficient to show the fibrosis and other injuries. The authors are suggested to perform some fibrosis-related protein expression (e.g., alpha-SMA or any other collagen types); and liver sections showing fibrosis by Masson’s trichrome or Sirius Red staining. 

Response: We greatly appreciate and thank the reviewer’s comment. According to the reviewer’s suggestion, Masson’s trichrome staining and semi-quantitation of collagen content (from ImageJ analyses) have been further performed to identify the effects of Asta-lipo against alcohol-induced liver fibrosis. The regarding information, findings and explanations have been added to Materials and Methods, Results and Discussion sections (Page 10, Line 24-231; Figure 7; Page 18, Line 333-334,351; Page 22, Line 531-541). 

                    (U)                                          (V)

Figure 7. (A-T) Masson’s Trichrome staining of the liver section in the experimental mice that received alcohol through two modes. Mice were given Asta-lipo or pure liposomes for 2 or 4 weeks prior to the 3-week 30% alcohol consumption (Groups C–F and K–N) or 3-week 30% alcohol administration plus a second dose of 30% alcohol (Groups G–J and O–R) as shown in Table 1. (U and V). Mice were treated with Asta-lipo (subfigure U) or pure liposomes (subfigure V) for 2 and 4 weeks and the collagen content in liver of each group was semi-quantitated by ImageJ. The measurement was repeated in four microscopic fields. (Group A, S and T were compared to Group B; Groups which received 2-week p.o and i.p. administration with Asta-lipo or pure liposomes were compared to Group S; Groups which received 4-week p.o and i.p. administration with Asta-lipo or pure liposomes were compared to Group T; *,#,+ P < 0.05, ##,++ P < 0.01 and *** P < 0.001,  by ANOVA).

2. The authors demonstrated that the increased serum ALT and AST as an indication of alcohol-induced liver disease which was further decreased after the Asta-lipo treatment. 

In reference to Kim et al., https://doi.org/10.1002/hep.22109, high ALT level is correlated with the severity of non-alcoholic fatty liver disease (NAFLD), not with alcohol-ic liver disease (ALD). This current study is focusing on ALD and its consequences on liver injury and fibrosis. How can an author discuss this point? The author should compare and discuss the ALD with NAFLD in detail at least in the discussion section explaining ALT and AST. 

Response: We greatly appreciate the reviewer’s comments. We reviewed many published literatures and summarized as follows: Previous studies in humans have shown that the Ratio of Aspartate Aminotransferase to Alanine Aminotransferase <1 suggest NASH, a ratio of 2 is strongly suggestive of alcoholic liver disease [1](Sorbi et al., 1999). In the mice p.o. or i.p. administered with Asta-lipo prior to 3-week 30% alcohol with or without feeding with second dose of 30% alcohol presented with AST:ALT values principally higher than 2, which is coherent with the previous conclusions by Sorbi et al. (1999). As to those mice administered with pure liposomes, Groups K, O , P and R showed the AST:ALT ratio less than 1, suggesting that subsequent administration with liposomes could be metabolic burdens under the liver conditions of 3-week 30% alcohol with or without feeding with second dose of 30% alcohol, leading to the AST:ALT ratio closer to NASH. However, longer-term (4 week) administration with liposomes prior to 3-week 30% alcohol with or without feeding with second dose of 30% alcohol, like the values shown in Groups M, N and Q, resulted in increased ratio of AST:ALT. This could be associate with that the longer halflife of mitochondrial AST released in response to alcohol and the coexistence of deficiency of pyridoxal-6-phosphate in alcoholics, which is a cofactor for the enzymatic activity of ALT [2](Diehl et al., 1984). 

Moreover, recent investigations using animal models have shown different serum biochemistry data to these previous findings in humans [3](Gao et al., 2017). Feeding male C57BL/6 mice with a Lieber-DeCarli diet containing 5% ethanol for 10 days, followed by a single dose of ethanol (5 g/kg body weight) by gavage, induces significant fatty liver and liver injury with peak serum levels of approximately 250 IU/L alanine aminotransferase and 420 IU/L aspartate aminotransferase 9 hours after gavage. Meanwhile, a second dose of 20% ethanol clearly led to increased ALT and AST [4](Ki et al., 2010). Therefore, it is not surprising that in our study the mice that received 30% ethanol orally for 21 days showed ALT around 150 IU/L coupled with AST about 250 IU/L.

All descriptions and changes have been added to Discussion section and related references have been cited and changed (Lines 400-430). 

Hopefully, these above-mentioned descriptions and revised changes can answer this comment and provide some contributions to readers.

Additionally, the ratio of AST: ALT is recommended to measure in all groups. 

Response: We greatly appreciate the reviewer’s comment. The ratio of AST/ALT of this study has been calculated and represented in Fig. 8E and 8F. And the statistical analyses and descriptions have been performed and included in the figure legend. 

Likewise, there are several non-invasive fibrosis markers as suggested by previous papers (DOI: 10.3748/wjg.v21.i39.11044; 10.1155/2014/357287). Why did the authors specifically perform only ALT and AST as a marker for liver injury and fibrosis in ALD? It needs to be clearly explained in the manuscript.

Response: We greatly appreciate the reviewer’s comment. Yes, we indeed agreed the reviewer’s comment on this point. However, the current study we extensively investigated and evaluated the therapeutic activities of Asta-lipo on the gross, histopathologic, serum biochemical changes in not only liver fibrosis but also in ALD. The obtained histopathologic observations and serum biochemical changes were evident and convincing. It is a great suggestion that to involve the other non-invasive fibrosis markers for ALD such as HA and PCIIINP (have been reported in [5] (Fallatah, 2014)), whereas their representativeness for extensive evaluation of therapeutic effects against ALD is probably not as strong as histopathologic and biochemical parameters. Thus, we designed and performed ALT and AST analyses only in this present study to evaluate the effect of Asta on the ALD (the other reason was due to the short of mice serum). Of course, the measurements of these non-invasive fibrosis markers will be included in our following related-studies. The related explanations have added in discussion section. 

3. The manuscript definitely lacks the molecular mechanisms; how Asta-lipo reduces the liver fibrosis in mice having alcoholic liver disease. The serum ALT and AST are just indicators of liver injury and not possesses a therapeutic mechanism to describe it. Is it possible to show the molecular mechanism/cellular signaling to make the manuscript smoother and easier to understand for all the readers?

Response: We greatly appreciate the reviewer’s comment. Yes, we indeed agreed that the molecular mechanism is a very important issue either to readers or researchers. However, the finding of molecular mechanisms on how Asta-lipo reduces the liver fibrosis are not the major topic or contents of this present study. Instead, the current study focused on the therapeutic and protective effects of liposomal Asta in mice with ALD. And the preliminary results from H&E stains, Masson’s trichrome stains and its semi-quantitation of collagen amount and biochemical analyses have been shown that the recovery of liver failure from ALD by treating with Asta-lipo. Anyway, we thank the reviewer’s great suggestion and the detailed molecular mechanisms of Asta-lipo against liver fibrosis and ALD will be investigated in our following study and hopefully will provide more contributions to readers. 

The lines 346-354 of discussion section explain some of the mechanisms for the protective effect of Asta in previous papers. The protective effects of Asta are already known by various mechanisms, however, in the current study, the authors have used Asta-lipo which is the main focus of the study. The protective mechanism of Asta-lipo is not explained in the discussion. So, the authors need to show some mechanisms by performing experiments to show the effect of Asta-lipo in alcohol-induced liver disease. 

Response: We greatly appreciate the reviewer’s comment. As mentioned in the introduction section, astaxanthin exhibits anti-inflammatory and free radical clearing action. Several studies have investigated the therapeutic or preventive effects of astaxanthin against cancer, inflammation or diet-induced obesity and hepatic steatosis [ref 6-8]. And many reports have indicated the protective mechanisms of Asta. However, unstable characteristic, low watery solubility and low bioavailability of Asta, all limited pure Asta applications in related-field. As for, we have successfully used liposomes or nanoparticles encapsulation techniques and analytical manners to assess the recovery of liver failure and repair from severe burns in the skin in vivo and publish journal papers [ref. 33, 35, 36]. Similarly, according to the extensive histopathologic observations (especially Masson’s trichrome staining for semi-quantitation of collagen) and related conclusions acquired, we revised the title to “Therapeutic and protective effects of liposomal encapsulation of astaxanthin in mice with alcoholic liver disease”, rather than mice with alcoholic liver fibrosis. Still the same, we thank the reviewer’s great suggestion and the detailed molecular mechanisms of Asta-lipo against liver fibrosis and ALD will be investigated in our following study and hopefully will provide more contributions to readers. 4. The explanation of Figure 7C and 7D is missing in the result section. It needs to be explained in the result section.

Response: We greatly appreciate the reviewer’s comment. The description and explanation of 7C and 7D (now is Figure 8C and 8D in revised manuscript) have been added to the result section (Line 280-299).

5. The model of group B (Fig 7C) is same as of group B (Fig 7A), but the data value looks varied, though they have measured ALT in both figures. Can the author explain about this point? Why there is variation in the error bars? The error bar of Fig 7A (group B) is around 190 mU/mL and the error bar of Fig 7C (group B) is around 175 mU/mL.

c We have checked the data in detail. The ALT value of group B has been consistent and the correct ALT value with its error bar has been added to Fig 8A and 8C. By the way, we have changed the scale of Y-axis to the same scale to avoid of confusing. 

6. In the discussion section, the authors have explained more about their results and less about the previously published literature. The majority of the results should be explained in the result section than in the discussion. Several points are repeated in the discussion from result sections. The authors are suggested to discuss more on the literature published in these areas, reduce the content which has already explained in the result sections and add more references to support the study.

Response: We greatly appreciate the reviewer’s comment. The content of explanation of results has been reduced in the discussion and more references have been quoted in the discussion to support the current study. And more results have been added and explained in the result section. (highlight in red color in revised manuscript) 

Other comments

1. What experiments were done for the ‘Serum biochemical data’ in Abstract portion Line 23? Need to explain the name of the experiments. 

Response: We greatly appreciate the reviewer’s comment. “Serum biochemical data” in Abstract have been appropriately revised. 

2. In the Introduction section Line 76-80, the sentence looks so long and difficult to catch the meaning. It’s better to split into two sentences.

Response: We greatly appreciate the reviewer’s comment. The sentence has been revised and divided into two sentences.

3. Under animal experimental design (Line 418), the authors have explained the treated dose of ethanol in a microliter, which is very unusual. It should be in mL/kg or µL/kg or g/kg (b.w). See some of the examples in these papers https://doi.org/10.1155/2017/7463571; 10.1016/j.jhep.2016.11.004, https://doi.org/10.3390/ijms17101616.

Response: We greatly appreciate the reviewer’s comment. The treated dose of ethanol has been revised as the explanation in the published papers by Zhou et al (2017) [9](Zhou et al. 2017).

4. Since the author mentioned about the statistical differences in body weight of different treatment groups (in result sections), the sign of statistical analysis is required in Figure 5 to show the differences among the groups and explanation in figure legends.

Response: We greatly appreciate the reviewer’s comment. The sign of statistical analysis and the differences among the groups and explanation have been added to Results section (also in figure legends).

5. How Asta-lipo or only liposomes were dissolved? What was the solvent used for oral and IP injection? Did the authors use enough control mice for the comparisons among the groups? What is the meaning of Normal Liver in Group A? The mice of this group were untreated (normal) or were treated with the vehicle?

Response: We greatly appreciate the reviewer’s comment.  Asta-lipo and pure liposomes were homogenized in sterile saline and then given to the mice via i.p. or p.o. Three control groups, including the mice without induction of alcoholic liver disease or any administration (Group A), the mice received 3-week alcohol feeding but without treatment for 2 weeks (Group S) and The mice received 3-week alcohol feeding but without treatment for 4 weeks (Group T) have been involved in this study. The mice in normal group are untreated. Also revised in Table1. 

6. In Figure 7, the explanation of figure legend is not enough. The author should explain the figures in more detail. What does A, B, C, D,……T means? It needs explanation in the figure legend too.

Response: We greatly appreciate the reviewer’s comment. The clear explanation of Fig. 7 has been added to its figure legend while the treatment conditions of some experimental groups has also been clarified in Table 1.

7. The statistical analysis of Figure 7 looks very confusing. To make easier for the readers, the authors should clearly mark and explain where they like to show the comparisons between the groups by different signs. They need to explain in the figure legends about the compared groups. One asterisk is seen near 350 unit (looks very unusual) in Figure 7C. What does that mean? They need to mention the ‘n’ number of mice in the figure legends also. 

Response: We greatly appreciate the reviewer’s comment. The comparisons between the groups by different signs in Fig. 7 have been reperformed and the explanation has been added to the figure legends. The “n” of each group has been clarified in the figure legends too.

8. What does p.o stand for? The full name of p.o is missing. What is the difference between Group B (Without p.o or i.p administration) and Group S and T (Without treatment for 2 weeks, without treatment for 4 weeks)?

Response: We greatly appreciate the reviewer’s comment. The full name of p.o has been added to the article. Group B stands for the mice liver prior to feeding 3-week 30% alcohol. Group S represents 2-week after the mice with 3-week alcohol feeding and Group T was 4-week after with 3-week alcohol feeding. Neither Group S nor Group T administered with any treatment after 3-week alcohol feeding. The information above has been added to the figure legends.

9. In Figure 6, the appearance of histological data is not clear. The magnification of the liver cells looks very small so that it makes difficulties to identify the inflammatory cells, fibrosis, necrosis or the repairment. Instead, high magnification images are suggested for the authors.

Response: We greatly appreciate the reviewer’s comment. The most lesions in the histopathologic figures were extensive, numerous and isolated. Therefore, 100× is the most appropriate magnification to clearly and entirely show the total lesions, degenerations and changes in the histopathologic figures of each group. This is also more convincing for the Asta-lipo treated groups to illustrate the minimization and improvement of the lesions compared to those without administration.

10. The author claims that the magnification of the images of Figure 6 is same i.e. 100x. However, the figure 6A shows different magnification image (50 µm) than other images (100 µm). The consistency is required throughout the figures and needs correction in the figure legend.

Response: We greatly appreciate the reviewer’s comment. The same magnification in Fig. 6A has been provided and corrected.

11. The schematic diagram of Figure 1C is not clear. Please add more description of the materials dose and name details on the figure as well as legends. In addition, if you show the H and E data on the alcohol-induced liver damage, then it is expected to show the image of the treated conditions.

Response: We greatly appreciate the reviewer’s comment. The materials dose, materials name and the instruments have been provided in Fig. (A). And the time scale of establishment of alcoholic disease and administration routes on the mouse model has been added. Figure 1 and the legends have been revised.

12. The protocol for histopathological analysis is not fully explained. “The tissues were then stained with eosin phloxine solution (Merck) for 30 s” cannot be the final steps of this staining protocol. The letter of Eosin should be capital in Line 432.

Response: We greatly appreciate the reviewer’s comment. The steps followed by eosin–phloxine solution staining have been added to the section and the letter of Eosin has been corrected.

13. In materials and methods section 4.7, the explanation of ALT and AST (from Line 438-446) is not suitable in this section. Better to keep in the introduction part. 

Response: We greatly appreciate the reviewer’s comment. The explanation of ALT and AST has been moved to Introduction and properly revised.

14. Some spelling mistakes were found such as radical-savaging (Line 462). Please check the grammatical and typos error throughout the paper.

Response: We greatly appreciate the reviewer’s comment. The spelling mistakes have been corrected and the grammatical and typos error throughout the paper have been rechecked.

Relaed References in comments answers:

1. Sorbi, D.; Boynton, J.; Lindor, K. D., The ratio of aspartate aminotransferase to alanine aminotransferase: potential value in differentiating nonalcoholic steatohepatitis from alcoholic liver disease. Am J Gastroenterol 1999, 94, (4), 1018-22.

2. Diehl, A. M.; Potter, J.; Boitnott, J.; Van Duyn, M. A.; Herlong, H. F.; Mezey, E., Relationship between pyridoxal 5'-phosphate deficiency and aminotransferase levels in alcoholic hepatitis. Gastroenterology 1984, 86, (4), 632-6.

3. Gao, B.; Xu, M. J.; Bertola, A.; Wang, H.; Zhou, Z.; Liangpunsakul, S., Animal Models of Alcoholic Liver Disease: Pathogenesis and Clinical Relevance. Gene Expr 2017, 17, (3), 173-186.

4. Ki, S. H.; Park, O.; Zheng, M.; Morales-Ibanez, O.; Kolls, J. K.; Bataller, R.; Gao, B., Interleukin-22 treatment ameliorates alcoholic liver injury in a murine model of chronic-binge ethanol feeding: role of signal transducer and activator of transcription 3. Hepatology 2010, 52, (4), 1291-300.

5. Fallatah, H. I., Noninvasive Biomarkers of Liver Fibrosis: An Overview. Advances in Hepatology 2014, 2014, 1-15.

6. Chang, S. H.; Huang, H. H.; Kang, P. L.; Wu, Y. C.; Chang, M. H.; Kuo, S. M., In vitro and in vivo study of the application of volvox spheres to co-culture vehicles in liver tissue engineering. Acta Biomater 2017, 63, 261-273.

7. Wu, Y. J.; Wu, Y. C.; Chen, I. F.; Wu, Y. L.; Chuang, C. W.; Huang, H. H.; Kuo, S. M., Reparative Effects of Astaxanthin-Hyaluronan Nanoaggregates against Retrorsine-CCl(4)-Induced Liver Fibrosis and Necrosis. Molecules 2018, 23, (4).

8. Wu, Y. C.; Wu, G. X.; Huang, H. H.; Kuo, S. M., Liposome-encapsulated farnesol accelerated tissue repair in third-degree burns on a rat model. Burns : journal of the International Society for Burn Injuries 2019, 45, (5), 1139-1151.

9. Zhou, T.; Zhang, Y. J.; Xu, D. P.; Wang, F.; Zhou, Y.; Zheng, J.; Li, Y.; Zhang, J. J.; Li, H. B., Protective Effects of Lemon Juice on Alcohol-Induced Liver Injury in Mice. Biomed Res Int 2017, 2017, 7463571.

Round 2

Reviewer 2 Report

Major concerns

1. The authors have shown the Masson trichrome staining of the livers of different groups, however, the resolution of the images is very weak. The background of the images looks very noisy which affects the quality of the data , therefore not so convincing also. I again suggest performing fibrosis-related protein expression to support the study. Also, the high-resolution images are expected in the final version.

2. The error bar and significant analysis are missing in Figure 8E and 8F.

Minor comments

3. As informed by authors, that the explanation of non-invasive fibrosis has been added in the discussion portion, I did not find the information regarding this issue. Could you highlight the line number of your explanation?

4.  The introduction of ALT and AST is required in the introduction section, but the explanation of the results is not necessary to include in the introduction. See Lines 71-88. Please revise carefully.

5. As suggested earlier, the spelling is not yet corrected. Please check the suggestions carefully.

Author Response

Major concerns

1.The authors have shown the Masson trichrome staining of the livers of different groups, however, the resolution of the images is very weak. The background of the images looks very noisy which affects the quality of the data, therefore not so convincing also. I again suggest performing fibrosis-related protein expression to support the study. Also, the high-resolution images are expected in the final version.

Response: We greatly appreciate the reviewer’s comment.

We have re-checked the quality and resolution of Masson trichrome staining in each group and these stains have been improved accordingly. The improved stains and semi-quantitation results hopefully are evident and will convince in this 2nd-revised version to demonstrate that administration with Asta-lipo is effective for amelioration of hepatic fibrosis in the mice model of alcoholic liver disease. And we especially thanks to this review comment that suggesting we perform more fibrosis-related protein expression to support this study. In fact, we are short of enough time (because we have to respond the reviewer’s comments in 10 days) and budget (research grant) to execute more protein expression experiment. Thus, we re-check and examine the H&E stains and Massion stains in detail and try to conclude the results correctly and accordingly (with all results obtained in this study). Yes, we will design more appropriate and more related experiments in the coming study and research grant.

Figure 7. (A-T) Masson’s Trichrome staining of the liver sections in the experimental mice that received alcohol through two modes. The observations were performed at 200× magnification (the image on bottom left site was 100× magnification). Mice were given Asta-lipo or pure liposomes for 2 or 4 weeks prior to the 3-week 30% alcohol consumption (Groups C–F and K–N) or 3-week 30% alcohol administration plus a second dose of 30% alcohol (Groups G–J and O–R)

2.The error bar and significant analysis are missing in Figure 8E and 8F.

Response: We greatly appreciate the reviewer’s comment.

The error bars and significant analysis have been added to Figure 8E and 8F.

Minor comments

1.As informed by authors, that the explanation of non-invasive fibrosis has been added in the discussion portion, I did not find the information regarding this issue. Could you highlight the line number of your explanation?

Response: We greatly appreciate the reviewer’s comment.

The explanation and discussion regarding the markers of non-invasive fibrosis in ALD has been added and highlighted in the discussion (Line 432 to Line442).

And we have added two references about this addition:

[24] .  Fallatah, I. H., Noninvasive Biomarkers of Liver Fibrosis: An Overview. Advances in Hepatology 2014, ID 357287, 1-15.

[25].     Lombardi, R.; Buzzetti, E.; Roccarina, D.; Tsochatzis, A. E., Non-invasive assessment of liver fibrosis in patients with alcoholic liver disease. World J Gastroenterol 2015, 21, (39), 11044-11052.

2.The introduction of ALT and AST is required in the introduction section, but the explanation of the results is not necessary to include in the introduction. See Lines 71-88. Please revise carefully.

Response: We greatly appreciate the reviewer’s comment. We have moved the related ALT and AST to the discussion section. The explanation of results is removed from the introduction section.

3. As suggested earlier, the spelling is not yet corrected. Please check the suggestions carefully.

Response: We greatly appreciate the reviewer’s comment. The spelling in the article has been carefully rechecked and revised by an English-spoken associate professor (the fourth co-author in this revised manuscript).

Round 3

Reviewer 2 Report

ijms-542710

The authors did not address all of the suggestions/comments provided by the reviewer, however, partially revised the suggestion. The most important parts have been addressed in the revised version. However, still minor revision is necessary which needs to revised carefully before accepting for the publication.

In Figure 6 figure legend, the citation of Figure A and B is missing. In Figure 7 figure legend also, the citation of Figure A and B is missing. What was the number of mice used in each group? How the change ratio of deposition calculated? How many mice samples were used in each group for histological analysis? Need to mention in the legends of Fig 6 and Fig 7. In Figure 8, the authors should also compare the treated groups with Group B (only alcohol feeding group). Why they have shown the statistics of group B only compared with Group A (control)?

Many abbreviations have not been defined. e.g., PCIIINP. Check the full manuscript again and add all of the missing explanations.

Result section 2.1 Line 91, the statistical analysis for Fig 1B is mentioned as mean±SD, but mentioned differently as SEM in figure legends. The author looks not sure about the method they used for analysis and during updating the manuscript

Author Response

The authors did not address all of the suggestions/comments provided by the reviewer, however, partially revised the suggestion. The most important parts have been addressed in the revised version. However, still minor revision is necessary which needs to revised carefully before accepting for the publication.

In Figure 6 figure legend, the citation of Figure A and B is missing. In Figure 7 figure legend also, the citation of Figure A and B is missing. What was the number of mice used in each group? How the change ratio of deposition calculated? How many mice samples were used in each group for histological analysis? Need to mention in the legends of Fig 6 and Fig 7.

Response: We greatly appreciate the reviewer’s comment:

The citation of subfigures A and B in Figures 6 and 7 have been added accordingly. The number of each group and the mice number used in each group is 4 (N=4). These have been added to the legends of Figures 6 and 7. The processes of change ratio of calculating deposition have also been added to the legends of Figures 6 and 7.

In Figure 8, the authors should also compare the treated groups with Group B (only alcohol feeding group). Why they have shown the statistics of group B only compared with Group A (control)?

Response: We greatly appreciate the reviewer’s comment:

The comparison between group B and the treated groups has been employed tin Figure 8. The reason why the comparison of Group A with group B was conducted is to confirm that the large increases in ALT/AST are consistent with those findings by histopathologic analysis.

Many abbreviations have not been defined. e.g., PCIIINP. Check the full manuscript again and add all of the missing explanations.

Response: We greatly appreciate the reviewer’s comment:

The full name of PCIIINP has been added and the other missing full names of abbreviations have been checked and added accordingly. By the way, we have added the full name of specific terms before the abbreviations in text.

Result section 2.1 Line 91, the statistical analysis for Fig 1B is mentioned as mean±SD, but mentioned differently as SEM in figure legends. The author looks not sure about the method they used for analysis and during updating the manuscript.

Responses: In result section 2.1 at Line 91, the statistical analysis (mean±SD) is for Fig. 1A not Fig. 1B. This typographical error has been corrected. In the rest of figure legends, mean±SEM was calculated and presented.
